# Seasonal discharge response to temperature-driven changes in evaporation and snow processes in the Rhine basin

Joost Buitink[1], Lieke A. Melsen[1], and Adriaan J. Teuling[1]

[1]Hydrology and Quantitative Water Management Group, Wageningen University, Wageningen, Netherlands

**Correspondence:** Ryan Teuling (ryan.teuling@wur.nl)

**Abstract.** This study analyses how temperature-driven changes in evaporation and snow processes influence the discharge in the Rhine basin. Using an efficient distributed hydrological model at high spatio-temporal resolution, we performed two experiments to understand how changes in temperature affect the discharge. In the first experiment, we compared two 10-year periods (1980s and 2010s) to determine how changes in discharge can be related to changes in evaporation, snowfall, melt from snow and ice, and precipitation. By simulating these periods, we can exchange the forcing components (evaporation, temperature for snowfall and melt, and precipitation), to quantify their individual and combined effects on the discharge. Around half of the observed changes could be explained by the changes induced by temperature effects on snowfall and melt (10%), temperature effects on evaporation (16%) and precipitation (19%), showing that temperature-driven changes in evaporation and snow (26%) are larger than the precipitation-driven changes (19%). The remaining 55% was driven by the interaction of these variables: e.g. the type of precipitation (interaction between temperature and precipitation) or the amount of generated runoff (interaction between evaporation and precipitation). In the second experiment we exclude the effect of precipitation, and run scenarios with realistically increased temperatures. These simulations show that discharge is generally expected to decrease due to the positive effect of temperature on (potential) evaporation. However, more liquid precipitation and different melt dynamics from snow and ice can slightly offset this reduction in discharge. Earlier snow melt leaves less snowpack available to melt during spring, when it historically melts, and amplifies the discharge reduction caused by the enhanced evaporation. These results are tested over a range of rooting depths. This study shows how the combined effects of temperature-driven changes affect discharge. With many basins around the world depending on meltwater, correct understanding of these changes and their interaction is vital.

## 1 Introduction

Over the last decades, global temperatures have increased considerably (Stocker et al., 2013). The resulting change in climate is generally expected to intensify the hydrological cycle, with more frequent and more severe hydrological extremes (Huntington, 2006). As increased temperatures affect water availability in large river systems in two important ways, it is vital to understand their effects and interactions. Firstly, higher temperatures affect the cryosphere: there will be less precipitation falling as snow and there is more energy available to enhance melt from snow and ice. As snow storages are depleted earlier in the year, it affects the timing of the snowmelt peak in the discharge signal (Jenicek and Ledvinka, 2020; Beniston et al., 2018; Baraer et al., 2012;

Huss, 2011; Hidalgo et al., 2009; Collins, 2008; Takala et al., 2009). Since meltwater from "water towers" is vital for billions of people (Viviroli et al., 2007), it is important to have a correct understanding of the expected changes in the cryosphere. Secondly, higher temperatures lead to increased potential evaporation rates, since a warmer atmosphere accommodates higher transport rates (Settele et al., 2015; Wild et al., 2013; Wang et al., 2010). With increased potential evaporation rates, discharge is expected to decrease. Several recent studies have investigated the discharge response to increased temperatures, and generally expect lower discharges resulting from increased evaporation and a shifted seasonality induced by the changed snow dynamics (Milly and Dunne, 2020; Mastrotheodoros et al., 2020; Rottler et al., 2020). However, the relative importance and the combined effect of evaporation and snow processes (including snowfall and -melt, and melt from glaciers) on discharge and its seasonal variability is currently not well understood.

Europe has experienced significant changes in evaporation, snow depth and streamflow over the last decades. For example, Teuling et al. (2019) showed that potential evaporation has increased by about 10% over the period 1960–2010. Their study shows that both changes in precipitation and evaporation had considerable effects on the streamflow. Additionally, a study by Fontrodona Bach et al. (2018) showed that snow depth decreased over the majority of Europe since the 1950s. Europe contains several major river basins of high socio-economic importance. One of these is the Rhine basin, with its headwaters originating from the Alpine region. This basin covers many different types of land cover: from glaciers to lowland areas. Several studies have already investigated the response of this basin under different climate scenarios (e.g., Stahl et al., 2016; Linde et al., 2010; Hurkmans et al., 2010; Pfister et al., 2004; Shabalova et al., 2003; Middelkoop et al., 2001). Yet, none of the studies have investigated the separate and combined response of evaporation and snow processes to rising temperatures.

Spatially distributed modelling becomes increasingly viable, due to the increased computational power, gains in model performance when adding spatial information (Comola et al., 2015; Lobligeois et al., 2014; Ruiz-Villanueva et al., 2012), and increased availability of high resolution data (e.g., Huuskonen et al., 2013; Cornes et al., 2018; Osnabrugge et al., 2017; C3S, 2017). However, the choice of spatial resolution can affect the model parameters (Melsen et al., 2016), and the sign of the simulated anomalies (Buitink et al., 2018). Besides, when finer spatial resolutions are used, the timestep should be reduced as well, as the space and time dimensions are linked (Blöschl and Sivapalan, 1995; Melsen et al., 2016). However, simulations ran at high spatial (and temporal) resolutions usually greatly increase computational demand (for example, the study by Mastrotheodoros et al. (2020) on a 250 m resolution required more than $6 \times 10^5$ CPU hours). This not only requires usage of high performance clusters, but also has an undesirable side-effect of increased power consumption (Loft, 2020). There is need for innovative hydrological models which can run on high spatio-temporal resolution without excessive computational demands, such as the recently developed dS2 model (Buitink et al., 2020).

This study investigates the hydrological response to temperature-driven changes in evaporation and snow processes (snowfall and -melt, and melt from glaciers). We test our main hypotheses that both seasonal changes in snow processes and enhanced evaporation will aggravate low flows, and that the changes will increase with temperature under realistic warming. We simulate the Rhine basin at high spatial (4 km) and temporal (1 hour) resolution using a calibrated version of the computationally efficient dS2 model, which is based on the simple dynamical systems approach (Kirchner, 2009; Teuling et al., 2010). The model was run for two decades, and for several scenarios with increased temperatures, to understand both historic changes and

potential changes in the future. The simulations performed at this rather unusually high temporal resolution ensures that diurnal variations—vital for evaporation and snow processes—are correctly represented. By separating the temperature-driven effects on evaporation and snow processes, we can understand and quantify the relative importance and interaction of each process.

## 2 Methods

### 2.1 Study area

This Rhine basin is one of the major basins situated in north-western Europe, covering several countries (see Fig. 1, also for typical climatic values). We focus on the basin upstream of the Netherlands, as this river is of vital importance for the country (e.g. agriculture, shipping). The basin includes part of the Alps, including several glaciers. Despite this being only a small fraction of the basin (as can be inferred from the digital elevation model in Fig. 1a), it has considerable effects on the discharge (Stahl et al., 2016). As a result, correct understanding of temperature driven changes is important to provide reliable discharge predictions. As every basin is unique in terms of hydrological, climatic and geological characteristics, the dependency of meltwater from snow and ice in discharge generation is not (Immerzeel et al., 2020). Therefore, we expect results from this study to give insight into the general hydrological response to higher temperatures.

### 2.2 Models and data

We used the computationally efficient distributed dS2 model (Buitink et al., 2020) to simulate discharge in the Rhine basin. The model is based on the simple dynamical systems approach (Kirchner, 2009), and is extended with snow and routing modules. As dS2 requires actual evaporation data as input, we ran a soil moisture model (BETA) prior to the rainfall runoff model to simulate the translation from potential evaporation (PET, calculated using the Penman-Monteith equation (Monteith, 1965)) to actual evaporation (AET). Since rootzone depth is an important yet highly uncertain parameter, we included simulations with rootzone depths ranging from 25 to 125 cm, with increments of 25 cm. All simulations are performed at a resolution of 4×4 km and at an hourly time step. The two models are explained in more detail below.

The input data were obtained from the ERA5 reanalysis dataset (C3S, 2017). This dataset is globally available on a $0.25 \times 0.25°$ resolution and at an hourly timestep from 1979 to present. ERA5 data were interpolated to the model grid using bilinear interpolation. We selected two periods with equal length based on the maximum distance between available decades of ERA5 data: 1980-1989 and 2009-2018, referred to as 1980s and 2010s, respectively.

Soil data were obtained from the European Soil Hydraulic Database (EU-SoilHydroGrids ver1.0, Tóth et al., 2017). As this dataset did not contain critical soil moisture content needed by the model to distinguish between water- and energy-limited evaporation regimes (Denissen et al., 2020), it was determined as the mean between wilting point and field capacity. The hygroscopic moisture content was calculated from the moisture retention curve based on Mualem-van Genuchten parameters at -10 MPa (Laio et al., 2001; Tóth et al., 2017). The clay content of the European Soil Hydraulic Database was used to calculate the pore size distribution ($b$) through a linear fit of the values found in Clapp and Hornberger (1978). For the depth of

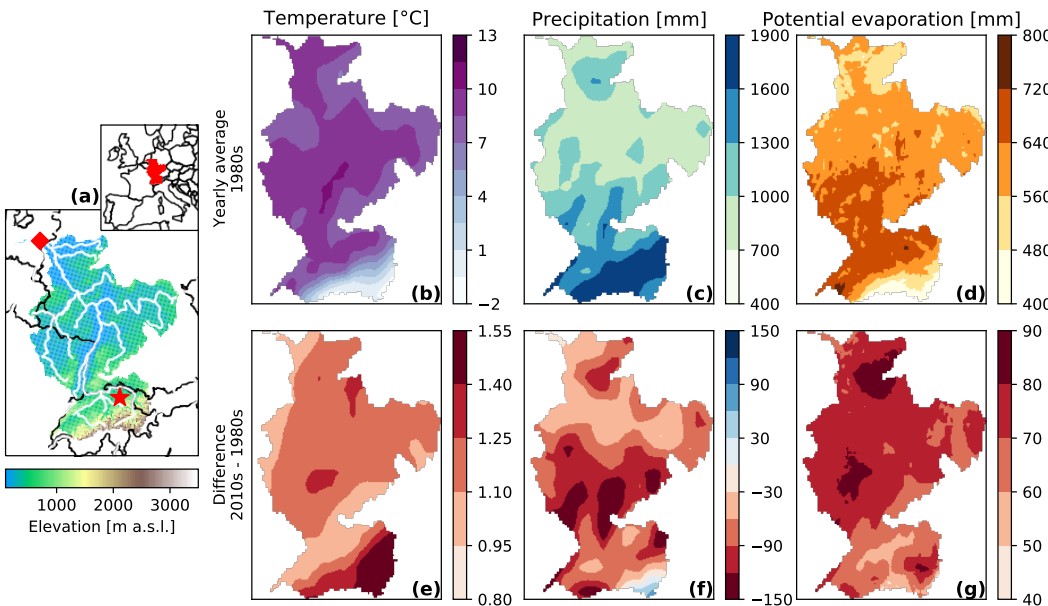

**Figure 1.** Digital elevation model of the Rhine, and hydro-meteorological changes between the 1980s and the 2010s. Panel a shows the simulation domain, with white lines indicating the main river branches. The red diamond indicates the location of the main outlet, and the red star indicates the location of the Rietholzbach research catchment using for validation. Inset shows the location of the basin within Europe. Top three panels show the yearly average values of the 1980s for temperature, precipitation and potential evaporation (b, c, d, respectively), and bottom three panels (e, f, g) show the differences between the 1980s and the 2010s.

the rootzone, we chose a depth of 75 cm, but also included simulations ranging from 25 to 125 cm with increments of 25 cm, to account for the uncertainty of this parameter. The potential evaporation input data was calculated using the Penman-Monteith equation (Monteith, 1965), based on ERA5 input data.

### 2.2.1 BETA

95

A simple soil moisture model (BETA, Beta EvapoTranspiration Adjustment) is used to preprocess the evaporation input. This model simulates the rootzone, and determines evaporation reduction based on the amount of water stored in the rootzone. Actual evaporation is assumed to be a function of the available soil moisture such that:

$$ET_{\text{actual}} = ET_{\text{potential}} \cdot \beta(\theta), \tag{1}$$

where $\beta$ represents the evaporation reduction parameter as a function of soil moisture $\theta$. $\beta$ is defined using three linear relations with $\theta$, based on Laio et al. (2001):

$$\beta(\theta) = \begin{cases} \beta_{\mathrm{w}} \frac{\theta - \theta_{\mathrm{h}}}{\theta_{\mathrm{w}} - \theta_{\mathrm{h}}} & \text{if } \theta \leq \theta_{\mathrm{w}} \\ \beta_{\mathrm{w}} + (1 - \beta_{\mathrm{w}}) \frac{\theta - \theta_{\mathrm{w}}}{\theta_{\mathrm{c}} - \theta_{\mathrm{w}}} & \text{if } \theta_{\mathrm{w}} \leq \theta \leq \theta_{\mathrm{c}} \\ 1 & \text{if } \theta_{\mathrm{c}} \leq \theta \leq \theta_{\mathrm{s}} \end{cases} \qquad (2)$$

where $\beta_{\mathrm{w}}$ represents the evaporation reduction factor at wilting point (set to 0.1), $\theta_{\mathrm{h}}$ represents the hydroscopic point, $\theta_{\mathrm{w}}$ the wilting point, $\theta_{\mathrm{c}}$ the critical soil moisture content, and $\theta_{\mathrm{s}}$ the saturated soil moisture content.

Leakage from the rootzone is calculated to simulate the vertical movement of water. This water is assumed to be gone from the rootzone, as we do not simulate a layer below the rootzone. The leakage is based on the unit-gradient assumption in combination with the Clapp and Hornberger (1978) model for unsaturated conductivity, integrated over a timestep $\Delta t$:

$$Q_{\mathrm{leakage}} = L\theta_{\mathrm{t}} - L\theta_{\mathrm{s}} \left[ \left( \frac{\theta_{\mathrm{t}}}{\theta_{\mathrm{s}}} \right)^{-2b-2} + \frac{(2b+2)k_{\mathrm{s}}\Delta t}{\theta_{\mathrm{s}}L} \right]^{-\frac{1}{2b+2}}, \qquad (3)$$

where $L$ represents the depth of the rootzone, $\theta_{\mathrm{t}}$ the soil moisture content at timestep $t$, $b$ the pore size distribution, and $k_{\mathrm{s}}$ the saturated conductivity. The value for $b$ is calculated through the clay fraction (CF), using a linear fit based on the values in Clapp and Hornberger (1978):

$$b = 13.52 \cdot \mathrm{CF} + 3.53. \qquad (4)$$

Finally, the water balance for the rootzone is defined as follows:

$$\theta_{\mathrm{t}+1} = \theta_{\mathrm{t}} + \Delta t (P_{\mathrm{rain}} + M_{\mathrm{snow}} - ET_{\mathrm{actual}} - Q_{\mathrm{leakage}}), \qquad (5)$$

where $P_{\mathrm{rain}}$ is the rate of rainfall at timestep $t$, $M_{\mathrm{snow}}$ the rate of snowmelt at timestep $t$, both are inferred the same way as in the dS2 model (see below and Buitink et al., 2020).

### 2.2.2 dS2

A conceptual rainfall-runoff model is used to simulate the discharge in the Rhine basin. The dS2 model (Buitink et al., 2020) is based on the simple dynamical systems approach, as proposed by Kirchner (2009). This approach is based on the assumption that discharge is a function of storage, such that changes in storage can be related to changes in discharge via a discharge sensitivity function:

$$Q = f(S), \qquad (6)$$

$$\frac{\mathrm{d}Q}{\mathrm{d}t} = \frac{\mathrm{d}Q}{\mathrm{d}S} \frac{\mathrm{d}S}{\mathrm{d}t} = \frac{\mathrm{d}Q}{\mathrm{d}S}(P - ET - Q), \qquad (7)$$

where $Q$ represents the discharge, $S$ the storage, $P$ and $ET$ the precipitation and actual evaporation, respectively, and $\frac{\mathrm{d}Q}{\mathrm{d}S}$ represent the discharge sensitivity to changes in storage, referred to as $g(Q)$. This concept has been successfully applied

and validated in several catchments across Europe (Kirchner, 2009; Teuling et al., 2010; Krier et al., 2012; Brauer et al., 2013; Melsen et al., 2014; Adamovic et al., 2015). Buitink et al. (2020) further developed the concept so it can be applied in a distributed way, to allow the simulation of larger catchments, while respecting the original scale of development. A new equation to better capture the typical shape of the g(Q) relation is proposed by Buitink et al. (2020), which contains three parameters:

$$g(Q) = e^{\alpha + \beta \ln(Q) + \gamma/Q}. \tag{8}$$

Additionally, the model has been extended with a snow module based on Teuling et al. (2010). The snow module is based on a degree-day method, adjusted to the hourly timestep. This snow module is conceptualized as follows (as defined by Buitink et al., 2020):

$$\frac{\mathrm{d}S_{\mathrm{snow}}}{\mathrm{d}t} = P_{\mathrm{snow}} - M_{\mathrm{snow}}, \tag{9}$$

$$P_{\mathrm{snow}} = \begin{cases} P_{\mathrm{total}} & \text{if } T <= T_0 \\ 0 & \text{if } T > T_0, \end{cases} \tag{10}$$

$$M_{\mathrm{snow}} = \begin{cases} \mathrm{ddf} \cdot (T - T_0) & \text{if } M_{\mathrm{snow}} \cdot \Delta t <= S_{\mathrm{snow}} \\ \frac{S_{\mathrm{snow}}}{\Delta t} & \text{if } M_{\mathrm{snow}} \cdot \Delta t > S_{\mathrm{snow}}, \end{cases} \tag{11}$$

where $S_{\mathrm{snow}}$ is the total snow storage in millimetres (mm), $P_{\mathrm{snow}}$ the precipitation falling as snow in millimetres per hour ($\mathrm{mm\ h^{-1}}$), $M_{\mathrm{snow}}$ is the snowmelt in millimetres per hour ($\mathrm{mm\ h^{-1}}$), T is the air temperature in degrees Celsius (°C), $T_0$ is the critical temperature for snowmelt in degrees Celsius (°C), ddf is the degree-day factor in millimetres per hour per degree Celsius ($\mathrm{mm\ h^{-1}\ °C^{-1}}$), and $\Delta t$ is the simulation time step in hours. We define separate melt factors for snow and glaciers. Partitioning of precipitation between liquid and solid precipitation (rain and snow) and melt is based on the critical temperature (set to 0°C). Routing of water is based on the width function (Kirkby, 1976), which means that lakes and other hydraulic structures are not explicitly simulated. More details on the routing module can be found in Buitink et al. (2020).

To calibrate dS2, we optimized the three discharge sensitivity parameters, the degree day factor for both snow and glacier pixels, and an evaporation correction factor. The evaporation correction factor is included to correct any bias errors in the forcing data. According to Boussinesq's theory of sloping aquifers (Rupp and Selker, 2006) and the results found in Karlsen et al. (2019), systems with higher slopes are expected to show higher discharge sensitivity values. Therefore, the discharge-sensitivity parameters were defined as a linear function of the slope of each pixel, based on the hypothesis that regions with steeper slopes show a more responsive storage-discharge relation than regions with gentle slopes. This resulted in two fitting parameters (slope and intersect) for each of the three discharge sensitivity parameters. Latin Hypercube sampling was used to gain parameter values evenly sampled across the possible parameter space. The period 2004–2008 was used for calibration. To ensure realistic model performance across the entire basin, the Kling-Gupta efficiency (KGE, Kling and Gupta, 2009) was calculated at 13 discharge measurement stations within the Rhine basin (see supplement for corresponding locations and

155 performance metrics). KGE values across all stations are averaged, and the parameters from the run with the best average KGE are selected. The resulting parameter values can be found in the supplement.

## 2.3 Experimental setup

We have split our analysis in two parts. Firstly, we compare two decades to understand the relative impact of each forcing variable. Secondly, we increase temperature values with 0.5 degree increments to understand how an increase in temperature

affects the hydrological response in the Rhine basin. We explain both experiments in more details below.

### 2.3.1 Validation

A thorough validation is required in order to ensure that models simulate the correct sign and magnitude of the trends (Melsen et al., 2018). Therefore, we validated dS2 on thee variables: discharge of the total catchment, and snow and evaporation dynamics. Additionally, validation of snow and evaporation is performed at two levels: local temporal validation with point

observations from the Rietholzbach research catchment (Seneviratne et al., 2012) in Switzerland (location is indicated with the red star in Fig. 1a), and spatial validation of evaporation and snow patterns using GLEAM (v3.3, Martens et al., 2017) and European Climate Assessment & Data Set (ECA&D, Tank et al., 2002; Fontrodona Bach et al., 2018). The Rietholzbach research catchment was selected as the data is available on a high temporal resolution (hourly), which matches our model setup. Snow observations from both the Rietholzbach and ECA&D are reported as snow depth, where dS2 simulates snow

water equivalent. In the graphs, we assume a transformation factor of 0.1 (where 10 mm of snow depth represents 1 mm of snow water equivalent), but always show both axes. The ECA&D dataset only contains point observations, and no stations in France are available. Due to data availability limitations of the Rietholzbach catchment, we had to resort to our calibration period. Since dS2 was only calibrated on discharge, this still can be interpreted as validation.

### 2.3.2 Forcing swap

In the first experiment, we aim to understand how each forcing variable can explain the resulting changes in discharge, and their relative importance. To perform this, we setup the experiment according to the conceptual overview presented in Fig. 2. The first two simulations are straightforward: using all forcing variables from either the 1980s or the 2010s to produce the corresponding discharge timeseries ("1980s" and "2010s"). In order to investigate how temperature influences evapotranspiration and snow processes separately, we perform model runs in which the total temperature change is separated into temperature effects on

evapotranspiration ("Changed $T_{evap}$") and snow processes ("Changed $T_{snow}$"). In addition, another run is performed with only changes in P ("Changed P"), so that these individual runs can be compared to a run where all changes in forcing are enabled ("2010s"). The "Changed $T_{evap}$" simulation changes the amount of evaporation which results from the new temperature time series. For the "Changed $T_{snow}$" simulation, the following snow processes are affected: type of precipitation (rain or snow), melt from snow and melt from glaciers. This experimental setup is, to our knowledge, new and has not yet been applied

in other studies. It allows to understand changes directly driven by a change in forcing, but quantifying combined changes

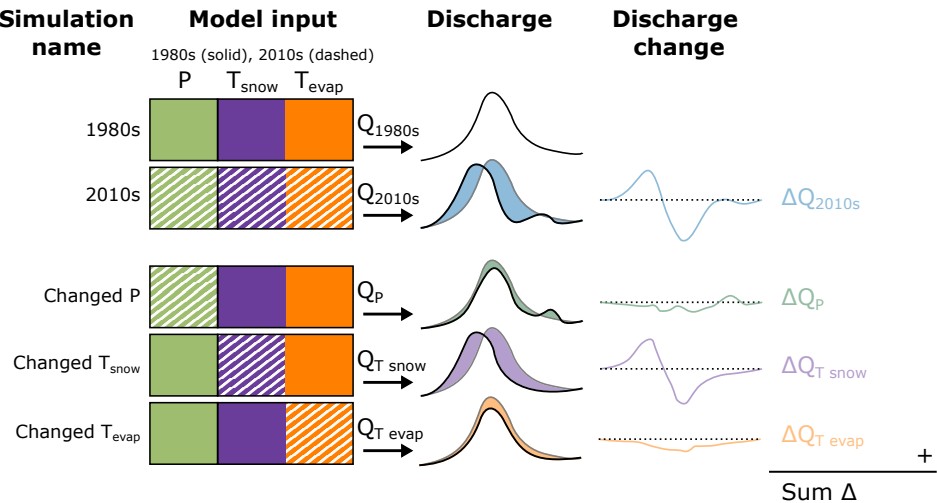

**Figure 2.** Conceptualization of the forcing swap experiment, showing the different simulations (rows) and steps in the analysis (columns). The different forcing variables are visualized as colored blocks, where the solid and dashed boxes indicate forcing data from the 1980s and the 2010s, respectively.

remains challenging (e.g. change in type of precipitation due to the interaction of precipitation and temperature). The resulting simulated discharge is compared to the 1980s run, to determine the discharge change. In this way, we can evaluate the relative impact of each forcing variable on the discharge.

We sum the discharge changes of the three forcing swapped runs, to obtain $\mathrm{Sum}\Delta$ (as timeseries):

$$\mathrm{Sum}\Delta = \Delta Q_P + \Delta Q_{T\ snow} + \Delta Q_{T\ evap} \tag{12}$$

where $\Delta Q_x$ represents the discharge difference of the forcing swapped simulations. We can use this $\mathrm{Sum}\Delta$ to study how well it explains the 2010s run, by comparing it to $\Delta Q_{2010s}$. We hypothesise that when $\mathrm{Sum}\Delta$ is equal to $\Delta Q_{2010s}$, the effect of the forcing is additive, and together explain all differences. We will refer to this as the direct effects. In the case of a discrepancy between $\mathrm{Sum}\Delta$ and $\Delta Q_{2010s}$, this can be attributed to interaction between the three forcing components. We will refer to this as indirect effects. For example, temperature and precipitation are linked as the type of precipitation (rain or snow) is dependent on temperature (see Eq. 10): a precipitation event in the 1980s would be falling as snow with the temperature of the 1980s ($T_{snow}^{1980s}$). When we swap $T_{snow}^{1980s}$ with $T_{snow}^{2010s}$, the same event could be classified as liquid rain leading to a direct runoff response, as opposed to the original snowfall event with snowmelt later in the year. This could also happen vice versa, or with swapped precipitation time series. The forcing-swapped simulations do not capture these interactions.

We define $\mathrm{Sum}\Delta$ to have explanatory value when it has the same sign as $\Delta Q_{2010s}$. We calculate the contribution of the direct effects ($\phi$) using the following equation:

$$\phi = \begin{cases} \frac{\min(\mathrm{Sum}\Delta, \Delta Q_{\mathrm{all}})}{\max(\mathrm{Sum}\Delta, \Delta Q_{\mathrm{all}})}, & \text{if } \mathrm{sign}(\mathrm{Sum}\Delta) = \mathrm{sign}(\Delta Q_{\mathrm{all}}) \\ 0, & \text{if } \mathrm{sign}(\mathrm{Sum}\Delta) \neq \mathrm{sign}(\Delta Q_{\mathrm{all}}) \end{cases} \tag{13}$$

This value can then be used to calculate the relative (direct) contribution of each forcing variable, using the following equation:

$$\phi_{\mathrm{x}} = \frac{\mathrm{abs}(\Delta Q_{\mathrm{x}})}{(\mathrm{abs}(\Delta Q_{\mathrm{P}}) + \mathrm{abs}(\Delta Q_{\mathrm{T\ snow}}) + \mathrm{abs}(\Delta Q_{\mathrm{T\ evap}}))} \cdot \phi \tag{14}$$

where $\Delta Q_{\mathrm{x}}$ should be replaced by $\Delta Q_{\mathrm{P}}$, $\Delta Q_{\mathrm{T\ snow}}$, or $\Delta Q_{\mathrm{T\ evap}}$.

### 2.3.3 Increased temperatures

In the second experiment, we raise temperature with $0.5°\mathrm{C}$ increments to understand how the basin responds to higher temperatures. We use the 1980s period as baseline, and increase the temperature until $2.5°\mathrm{C}$, to match realistic temperature projections. Similar to the previous experiment, we separate the effects of temperature on evaporation and snow processes: an increase in $\mathrm{T_{evap}}$ influences the resulting evaporation, where an increase in $\mathrm{T_{snow}}$ affects the type of precipitation and the melt from snow and glaciers. By separating these effects, we can understand their relative importance for each temperature increase. This is a simplified approach, as a recent study by van der Wiel and Bintanja (2021) showed that a warming climate not only affects the mean, but also the variability. The latter is not capture in our approach.

## 3   Results

### 3.1   Forcing comparison and validation

A first comparison of average temperature, precipitation and potential evaporation reveals considerable differences between the two periods (Fig. 1). Over the entire Rhine basin, yearly average temperature has increased with more than $1°$ C, from $8.1°$ C to $9.3°$ C between 1980s and 2010s. Largest differences are found in the eastern Alps, where average temperature has risen by $1.5°$ C (Fig. 1b, e). Average precipitation is lower in the 2010s over the majority of the Rhine basin, with the yearly average precipitation sums decreasing from 1146 mm to 1066 mm (Fig. 1c, f). Spatial differences in precipitation are, however, less homogeneous over the basin than the changes in temperature and potential evaporation. As a result of the increased temperatures, average potential evaporation also substantially increased from 607 mm to 678 mm from the 1980s to the 2010s, with the largest increases occurring in the northern parts of the basin (Fig. 1d, g).

Discharge validation (Fig. 3a) shows that dS2 simulates the discharge with high KGE values in both periods. Panel b shows how the average discharge differs between the two periods, with lower discharges in the 2010s for the majority of the year. Discharge during the 2010s does not show as high discharge values in June, and shows lower discharge values occurring later in the year. Kling-Gupta efficiencies for each period and several stations within the basin can be found in the supplementary information (Table S1).

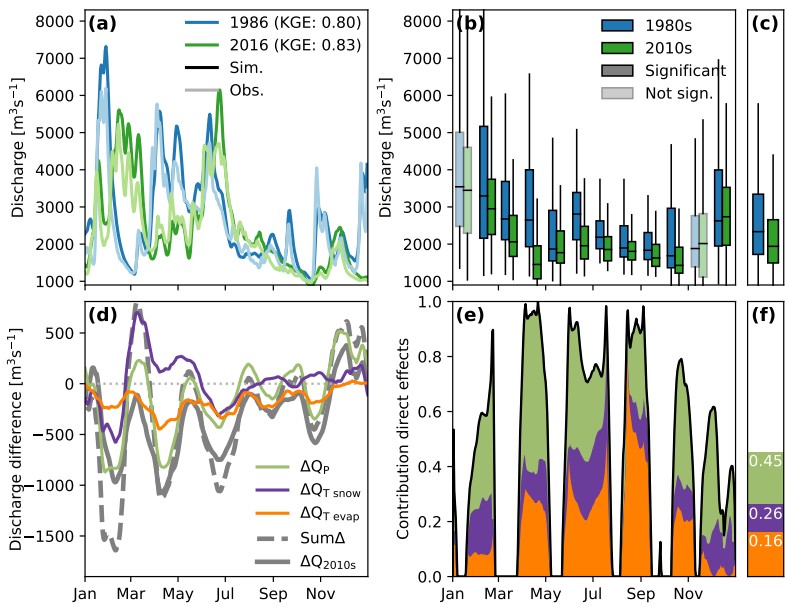

**Figure 3.** Attribution of discharge changes between the 1980s and the 2010s. Panel a compares simulated (dark colored) with observed (light colored) discharge values for one representative year in each period. Panel b and c compare the monthly and yearly (respectively) simulated discharge values between the two periods, where full coloured boxes are significantly different (p<0.05, based on independent t-test). Panel d shows the difference of model simulations where one of the forcing data has been swapped with the time series from the 2010s. Panel e shows the contribution of direct effects (black line), and the contribution of each forcing variable, to the total change between the two periods. Overall mean values of the timeseries in panel e are presented in panel f.

For the validation with data from the Rietholzbach catchment, we compare simulated actual evaporation with observed
evaporation from a lysimeter, and compare simulated snow storage with observed snow height measurements in Fig. 4. Both variables are correctly represented, and show similar variability as the observations, even at hourly timescale. The simulated evaporation generally shows a smoother signal than the observations. Snow storage shows a very similar pattern. It has to be noted that snow height observations cannot be directly converted into snow water equivalent, due to e.g. compaction. Yet, dS2 simulates melt and snowfall at moments corresponding with observations, as is confirmed by the contingency table in panel b.
Given that dS2 is not calibrated on these variables, and the difference in spatial scale of the input data, this shows that dS2 is able to correctly simulate evaporation and snow processes.

This is confirmed in the spatial validation, where we compare the actual evaporation results from the BETA model with results from GLEAM (Martens et al., 2017). We see some deviations in terms of magnitude, where BETA simulates slightly higher values than GLEAM. However, despite the differences in spatial resolution, the patterns are well represented in BETA:
higher values in the southern region of the basin, with lower values in the middle/northern regions. The simulated snow storage is compared with observations from the ECA&D dataset, provided by Fontrodona Bach et al. (2018). It should be noted that the observations are measured as snow height, while dS2 simulates snow water equivalent. The figure shows that dS2 simulates

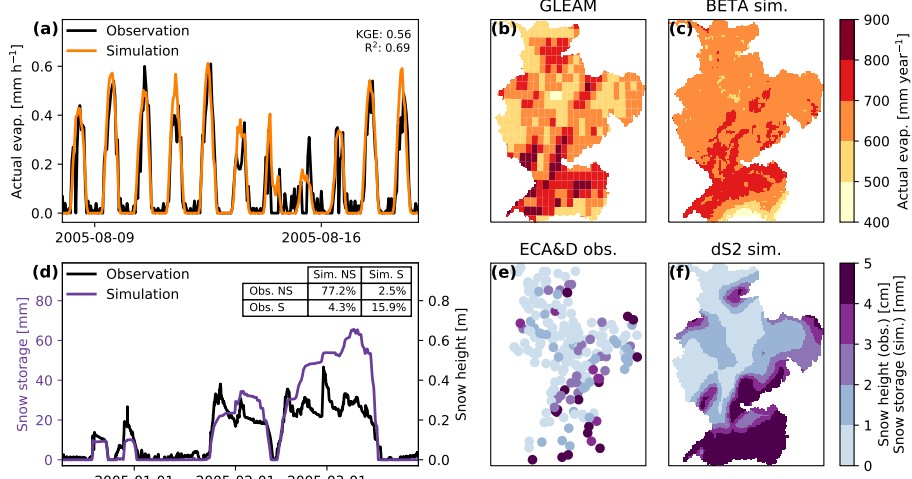

**Figure 4.** Temporal and spatial validation of evaporation rates (a, b, c) and snow storage (d, e, f). Panels a and d show validation with observations from the Rietholzbach research catchment (location can be found in Fig. 1a), with the table in panel d showing the contingency table with the percentage of occurrences with snow (S) and with no snow (NS). Note that panel d has two y axes: snow storage for the simulation and snow height for the observations. Panels b and c show the annual mean actual evaporation of 2005 as determined with both GLEAM and BETA (with a rooting depth of 75 cm). Panels e and f show annual mean snow heights of 2005, as observed in the ECA&D dataset (snow height in cm), and simulated with dS2 (snow storage in mm).

snow cover with a similar pattern as is observed: with high values in the Alps and southeastern region, and low values in the middle and northern parts of the basin.

## 3.2 Forcing swap

Investigating the differences between the "forcing-swapped" runs gives insight on how each variable affects the discharge (see Fig. 3d, e, and f). Changing the precipitation ($\Delta Q_P$) has substantial effects on the discharge (Fig. 3d), swinging from large negative discharge differences to positive differences. This is not unexpected, as precipitation is the factor controlling water input into the basin. Changing only the temperature affecting snow processes ($\Delta Q_{\mathrm{Tsnow}}$, including type of precipitation, and melt from snow and glaciers) shows discharge differences mostly in the first half of the year. The reduction of discharge in January and February is caused by overall higher temperatures: less precipitation has fallen as snow in the preceding months (as inferred from the increased discharge at the end of the year), leading to less snowmelt in January and February. From March to May, this simulation shows higher discharge values resulting from a more direct discharge response due to more rain (instead of snow), and enhanced melt from the glaciers. From July onwards, discharge values converge back to the original 1980s simulation, indicating that the discharge regime becomes less dominated by melt from snow and ice. The simulation with evaporation from the 2010s ($\Delta Q_{\mathrm{Tevap}}$) shows a discharge reduction over the entire year. The higher PET leads to higher actual evaporation, decreasing the discharge.

The contribution of direct effects in Fig. 3e-f gives an indication on the amount of interaction between the three forcing variables. Values close to 1 indicate that there is little interaction, as the sum of the differences is able to explain all changes. The contribution of direct effects is lowest during March and around October. During these periods, the storage conditions of the basin largely control the discharge response, either through snow storage or water available to generate runoff. Around March, changes in the available snow storage are the result of interactions between temperature and precipitation. Around October, discharge is controlled by water that is available for runoff generation, which is controlled by interactions of precipitation and evaporation. Additionally, since the response of a pixel is a function of its storage (through the simple dynamical systems approach), this also affects the runoff response. These interactions of forcing variables cannot be captured by simply combining the individual discharge responses, hence the relatively large contribution of indirect forcing effects during these periods. When taking the averages of the values in Fig. 3e, the results show (Fig. 3f) that, overall, it is possible to explain almost half of the 2010s discharge scenario by the direct forcing effects. The temperature effects of evaporation and snow (0.16 and 0.10, respectively, totalling to 0.26) are just as important as the changes induced by differences in precipitation (0.19), yet due to the large role of interactions, no more than 45% can be explained using this simple addition.

## 3.3 Increased temperatures

In the second experiment, we investigate the role of temperature increases on changes in discharge. Higher temperatures affect the hydrological cycle either through evaporation, snow processes (snowfall, and melt from snow and ice), or a combined effect of the two. Using the dS2 model, separate simulations of temperature effects on evaporation ($Q_{Tevap}$), snow ($Q_{Tsnow}$) and their combined effect ($Q_{Tboth}$) allow us to understand which variable is causing the main changes. These time series are presented in Fig. 5a, including the 1980s run as reference. Throughout the year, we see a near-constant reduction in discharge, without clear seasonal-patterns. To investigate this further, we highlighted three periods representing typical discharge regimes: high discharge during January and February, the meltwater peak during May and June, and low discharge during September and October. For each of these periods, the change in discharge shows a roughly linear relation with temperature increase. This is in line with our hypothesis, that higher temperatures will lead to larger differences in discharges. The resulting near-linear relation is interesting, as both snow and evaporation processes are threshold processes, through their relation with temperature and soil moisture, respectively. We expect to see a more non-linear response to temperature when reaching more hydrological extremes.

Surprisingly, for the periods during January-February and September-October (Fig. 5b and d), the modified snow run shows behaviour opposite to both the modified evaporation run and the combined run. In these cases, the increased discharge as result of a change in snow processes (more liquid precipitation, and more meltwater production from the glaciers) slightly offsets the negative discharge change induced by the increased evaporation. This ensures that the combined reduction in discharge is less severe than when only the reduction induced by evaporation is considered. However, during May-June (Fig. 5c), both evaporation and snow processes show a negative discharge change, enhancing the combined negative change in discharge. During this period, less snow was available to melt, leading to a reduction in discharge. As a result, the discharge of the combined run shows an even larger reduction in discharge, where even the peak during June from the 1980s has been largely

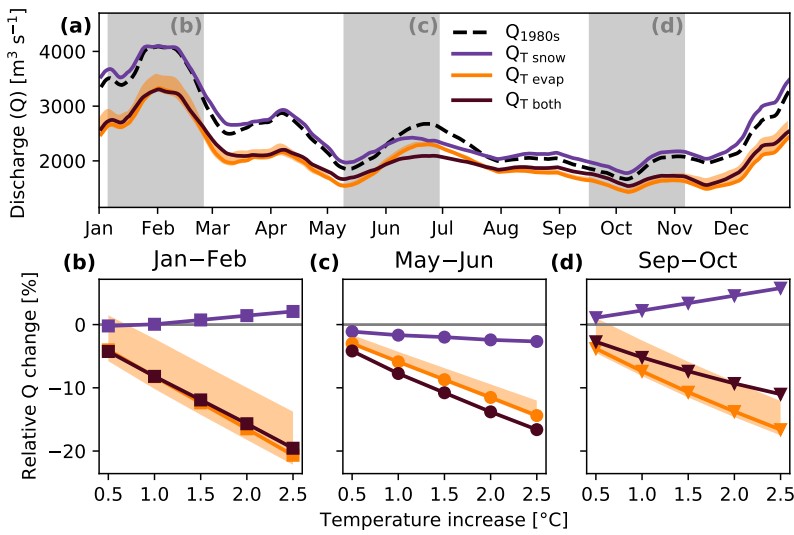

**Figure 5.** Discharge sensitivity to temperature increase. Panel a shows the yearly average discharge under a 2.5°C increase, and panel b shows changes during typical discharge events with stepwise temperature increases. Typical discharge periods highlighted in panel a match the periods used to compare the mean discharges in panels b, c, and d. Shaded orange areas indicate the uncertainty induced by effective rooting depth (25–125 mm), where higher discharges match shallower depths and vice versa.

diminished (Fig. 5a). The exact response of type of precipitation, snowdepth, -cover, -melt and melt from glaciers for each temperature increase can be found in the supplement. Substantial influence of rooting depth on the evaporation simulation is visible (shaded orange areas in Fig. 5), yet the trend direction with increasing temperatures remains equal. Shallower rooting
depth values induce more soil moisture stress since less water is available, leading to higher average discharges.

To understand the cause of these changes, the change in generated runoff per model pixel is shown in Fig. 6a. This figure shows that the majority of the basin produces less runoff for all three periods. Only the southern regions of the basin show a different response. During January and February, these regions produced more runoff, resulting from the increased snowmelt and increased liquid precipitation. In the other periods, only a few pixels produced more runoff. These pixels correspond to
the glaciers in the Alps, which produced more meltwater resulting from the increased temperatures, and explain the positive discharge change in Fig. 5d.

In Fig. 6b, the fraction of the basin that is dominated by one of these three options is plotted against the relative change in mean discharge for each period. As expected, the majority of the basin is mainly influenced by evaporation (84–94%). As a result, the mean discharge is reduced by ±17%. Contrasting, a limited fraction of the basin (1–6%) is mainly influenced by
snow processes, yet still has a considerable effect on the mean discharge: varying between −3% and 6%, depending on the period. Pixels in the basin where neither snow or evaporation appeared dominant take only a very small fraction of the basin (<1%). Generally, these regions are at the transition between snow dominated and evaporation dominated regions. Overall,

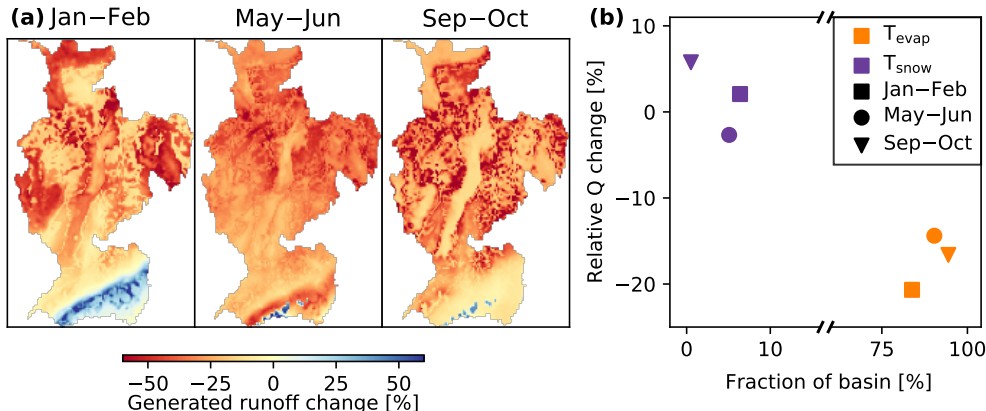

**Figure 6.** Spatial differences in the Rhine basin under the +2.5°C scenario. Panel a shows the differences in generated runoff for the three periods highlighted in 5a. Panel b shows the fraction of the basin where 80% of the changes could be explained by either evaporation or snowmelt, and the average discharge change corresponding to each process.

the change induced by $T_{snow}$—despite the small contributing area—substantially affects the discharge. More details on the response for each temperature increase can be found in the supplement.

## 4  Discussion

We compared two periods of 10 years to investigate the relative importance of changes in temperature, evaporation and precipitation. Over these periods of 10 years, most interannual variability is averaged out, allowing us to objectively investigate the effect of different temperatures on the hydrological response. However, decadal variation remains present, but due to the length of these periods, it is not possible to fully contribute these changes to a change in climate.

When validating the model with data from the Rietholzbach, the simulated evaporation in the Rietholzbach showed a smoother signal than the observations. Deviations between the observations and simulations could be caused by the relatively coarse ERA5 data. Especially in the spatially heterogeneous Alps, a single ERA5 pixel could miss some local variations (such as variations in cloud cover, radiation, wind and/or temperature), causing a smoother signal in the simulated evaporation. For the spatial evaporation validation with GLEAM, it should be noted that this is not a true validation, since GLEAM is not a completely independent observational dataset. Despite this, GLEAM is often considered as a reference for spatio-temporal validation of ET. Additionally, BETA uses a single rooting depth value for the Rhine basin.

The choice of spatial model resolution is a balance between data availability, computational time and underlying modelling concept. Here we selected a resolution of 4×4 km, so we can use the ERA5 forcing data with bilinear resampling (without adding more degrees of freedom, uncertainty, and potential errors), have short runtimes (simulating 10 years including all I/O operations takes just over 5 minutes on a normal desktop), and apply the model at it's proven spatio-temporal scale ($\pm 10$ km$^2$

at hourly timestep). Contrasting, the study by Mastrotheodoros et al. (2020) used a much finer spatial resolution, but at the cost of enormous CPU times.

Several other recent studies have investigated the hydrological response to increased temperatures via either changes in evaporation and/or snow processes (snowfall, and melt from snow and ice). Below, we compare our results with the results found in three of these studies. Firstly, the study by Rottler et al. (2020) showed a decrease in runoff seasonality in rivers fed by melt water from snow, over the period 1869–2016. They found higher discharges during winter and spring, and lower discharges during summer and autumn. The authors conclude that reservoir constructions in these snow-dominated rivers is likely to cause this redistribution of discharge. However, our temperature experiment shows a similar change in discharge: with higher discharges in winter and spring, and lower discharges during summer. As the dS2 model does not include dams and other reservoirs, this signal can be attributed to a change in discharge production. Secondly, a recent study investigating the response of several basins in Czechia to changes in snowmelt, concluded that snowmelt started earlier in the year, which also reduced summer low flows via baseflow (Jenicek and Ledvinka, 2020). However, our study shows that low flows during September/October actually increased when only snow processes are considered. This can be explained by that fact that the Rhine basin includes glaciers, which produce more meltwater with higher temperatures (assuming the glacier is thick enough to facilitate this melt). This increase in meltwater resulted in higher discharge volumes during summer. Thirdly, Milly and Dunne (2020) showed a reduction of discharge in the Colorado river basin, and conclude that this is driven by increased evaporation. This increase of evaporation is attributed to a reduction in snow cover, and hence a decreased albedo. Despite the different basin, our study supports the conclusion that increased temperatures reduce discharge through both changes in snow processes, and increased evaporation. Our model does not account for changes in albedo, but does allow areas previously covered in snow to evaporate water. And while differences in climate zone of the Colorado and Rhine basins make it challenging to compare the absolute numbers, the sign of the trend is equal.

Musselman et al. (2017) concluded a lowering of snowmelt rates due to a shift of the melt season towards a period with lower available energy (spring in stead of summer). They simulated the snowpack with a more complex energy balance, rather than our degree-day method. The dS2 model does not include the radiation driven changes, but does simulate that snowmelt occurs earlier in the year. This means that, under increased temperature scenarios, snowmelt occurs on days where previously no melt was possible. Additionally, dS2 does simulate earlier depletion of the snowpack, which also reduces snowmelt rates. Despite our different aim and approach, we believe that our study supports the findings of Musselman et al. (2017).

The glaciers used in our model are fixed in space, and no growing or shrinking of glaciers is simulated. While the approach is common for shorter time scale studies such as ours (van Tiel et al., 2018), it limits the interpretation of our results several decades into the future. The study by Lutz et al. (2014) included a glacier mass balance, and showed that melt from glaciers increased before the glaciers eventually disappear. Without glaciers, increases in drought severity are expected, as less meltwater is produced during the summers (van Tiel et al., 2018; Huss, 2011). After 2050, substantial changes in summer flow resulting from the reduction in glacierized area are expected (Huss, 2011). Until this period, we expect our results to be representative.

When comparing our results with results from studies perform in the Rhine basin (e.g., Linde et al., 2010; Hurkmans et al., 2010; Pfister et al., 2004; Shabalova et al., 2003; Middelkoop et al., 2001), we see similar results. These studies focussed

mainly on understanding and/or projecting the Rhine discharge under climate scenarios. Yet all these studies agree that the snowmelt peak will occur earlier in the year, and that the basin is expected to transition from a mixed rain/snow-fed river to a mostly rain-fed river. Additionally, all studies agree to expect higher evaporation rates, further reducing the discharge. This is all in line with our study, despite the fact that we did not investigate changes in precipitation in our temperature scenarios. Furthermore, the Rhine basin contains several hydraulic control measures, which are currently not represented in our model structure. Despite this, we still reach good model performance, suggesting that these structures currently do not have a very large influence on the discharge dynamics at the basin outlet. In the future, however, the management schemes and number of structures can be altered to accommodate the changes in the hydrological cycle. As these changes are a large unknown, we decided to only focus on the natural hydrological response.

## 5   Conclusions

Temperature, evaporation and precipitation substantially changed from the 1980s to the 2010s in the Rhine basin, reflecting changes that are typical for many larger basins around the world. In the 2010s, basin average temperature was more than 1°C higher, potential evaporation was almost 70 mm higher, and precipitation decreased with 80 mm. Discharge between these two periods was significantly different for 10 out of 12 months. Each individual forcing variable can partly explain these discharge differences: 10% can be explained by the changed snowfall and melt dynamics, 16% is explained by the changed evaporation, 19% by the changed precipitation; leaving 55% to be explained by the interaction of these variables. As differences in evaporation, snowfall and melt are driven by changes in temperature, the temperature effect is larger (26%) than the changes induced by changes in precipitation (19%).

With higher temperatures, discharge is expected to decrease, resulting from the postive effect of temperature on (potential) evaporation. However, snow processes (more liquid precipitation and enhanced melt from glaciers) can partially offset the negative change in discharge during the lowflows in September-October, which was contrary to our expectations. The discharge response during May-July matches our hypothesis that both changes in snow processes and evaporation enhance the reduction in discharge. This is a result of the combined effect of enhanced evaporation, and a reduction in snowpack leading to less snow melt.

This study focusses on the Rhine basin, yet these results can provide insight for the many different basins around the globe, which also depend on both rain- and snowfall. With higher temperatures, changes in snow processes slightly offset the discharge reduction from enhanced evaporation over the majority of the year. However, the season where runoff generation is reduced due to smaller snow storages (and potentially smaller glaciers) should be identified in each basin, as this part of the year is impacted the most. Many regions rely on "water towers" for their year-round water availability (Immerzeel et al., 2020), where the mountainous regions cover varying fractions of the basin. In many basins, more of the discharge originates from these water towers than in the Rhine basin, amplifying our results. Here, higher temperatures would likely imply even stronger negative amplitudes in discharge trends during the melt season. Enhanced melt from glaciers and a shift from snow to rain can partially

offset the negative change in discharge caused by the increased evaporation, but can enhance the negative change when snow storages are eventually depleted earlier in the year.

*Code and data availability.* Model code and information is available at Buitink et al. (2020). Forcing data was obtained from C3S (2017). Soil data was obtained from Tóth et al. (2017).

*Author contributions.* JB and AJT designed the study. JB performed the model simulations and analyses, and wrote the manuscript with contributions from LAM and AJT.

*Competing interests.* The authors declare that they have no conflict of interest.

*Acknowledgements.* We would like to thank Christoph Brühl and Doke Schoonhoven for their work during their MSc. thesis, which set the foundation for this study.

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
