# Peer review of "Seasonal discharge response to temperature-driven changes in evaporation and snow processes in the Rhine basin"

_Earth System Dynamics, 2020_

## Referee Comment (RC1) · Anonymous Referee #1 · 26 Oct 2020

General comment

Buitnik et al. simulate the Rhine River Basin for two 10-year time slices (1980s and 2000s) using the dS2 model and ERA5 data (0.25x0.25). An additional soil moisture model was added in order to attain values of actual evapotranspiration. ERA5 data was interpolated to a 4x4 km grid using bilinear interpolation. They assess changes in discharge between the two time slices and attribute differences in runoff to differences in precipitation, evapotranspiration and snowmelt. In general, it is an interesting study and has the potential to become a valuable contribution to hydrological research. However, I see several major issues regarding the analysis and the text that need to be

addressed before it can be considered for publication.

Major comments

Comment 1: Structure text

One major issue I see is that sections in the manuscript are mixed up or even missing. Often, model and method description are located in the result section. The results section is mixed with the discussion. In general, the manuscript lacks important details. I think a more detailed method description, where each step of the analysis conducted is described, needs to be added. Furthermore, model set-up and model components need to be described better. This is crucial to understand you experiments. Please provide more information on the dS2 model and the snow routine. An additional section 'study area and data' could be good to better introduce the Rhine Basin. In my opinion, large parts of the supplementary material can be moved into the actual manuscript.

Comment 2: Swapping variables

I still have troubles to understand your approach of swapping forcing variables in order to attribute changes. You try to asses the contribution of the three factors: 1) changes in precipitation, 2) changes in snowmelt due to changes in temperature and 3) changes in evapotranspiration due to changes in temperature. If I understand correctly, you swap individual forcing variables (temperature or precipitation, respectively) between the two 10-year time slices. You run the 1980s with temperature data from the 2000s, for example. So the temperature in the first week of August 2005 becomes the temperature in the first week of 1985, right?

This seems to be a very rough approach and I am not sure if this is a good idea. Also the performance of this approach seems quite poor. For parts of the year you can not at all explain variations or even expect an opposite trend (in February, for example, strong overestimation and in March even an opposite trend). The three factors you investigate do not really explain the variations in discharge, I think.

All variations that you can not explain, you attribute to 'interactions'. To me, these 'interactions' are not clear. I do not know what you have in mind here. Can you explain more detailed?

It looks like that a model only using the factor 'changes in precipitation' alone would explain changes in discharge better than the approach with the three factors (Fig. 2e). Please check.

What about changes in evapotranspiration due to changes in precipitation? What about changes in snowmelt due to changes in precipitation?

Comment 3: Attribution

In the abstract you write that variation can be 'explained by the changes induced by snow (11%), evaporation (19%) and precipitation (18%), while 52% was driven by combination of these variables." This bases on results presented in Fig. 2 panels e, f and g, I assume. How do you calculate those percentage values? In February and April, for example, you have a negative variation in discharge between the two 10-year slices of around 1000 m$^3$/s. This variations can be explained by variation in rainfall to 90-100%, right?

Comment 4: Evapotranspiration

Are you using evaporation or evaporation + transpiration = evapotranspiration? You mention the Penman-Monteith equation to get potential evaporation. Isn't it reference evapotranspiration you get? Do you calculate this? Or is it ERA5 data directly? You show a map of the potential evaporation (Fig. 1c). Can you also show a map of the calculated actual evaporation? Please give more information on the soil hydraulic data used. If I understand correctly, you assume one value of the rootzone depth for the entire basin? Is this value also constant over time? What about vegetation cover?

Comment 5: Temperature 'scenarios'

In my opinion, the different approaches you use need to be explained better. Your

swapping approach and the temperature 'scenarios'. Please explain in the method section. In those temperature 'scenarios', you simple add e.g. 2 °C to the hourly temperature time series? This again seems a very simple approach. In what way do those increases influence your model components and does this reflect 'real-world' processes? In Fig. 5 b your results indicate that increasing temperatures will rise evaporation in winter, in turn, decreasing discharges. According to my knowledge, evapotranspiration does barely play any role in winter (low radiation, low temperatures, no plants). All studies I know hint at an increase in discharge in winter, e.g. due to liquid instead of solid precipitation. Can you check this and maybe show your simulated evapotranspiration values for winter for this temperature 'scenarios'? Your model shows a linear response of evapotranspiration to temperature. How is evapotranspiration affected by changes in temperatures in the first place? Isn't it an input to your model?

Comment 6: Validation, Calibration

Why do you use two point measurements of this small pre-Alpine catchment to validate? Isn't there hundreds of snow gauges? Where is this snow gauge exactly? At what elevation? You compare the simulated snow in your 4x4km cell with the point measurement? Why do you simulate in an hourly resolution? Why not daily? For the routing? Can't you use daily data for calibration and validation (even if you initially run hourly)? At the end you aggregate to monthly values anyway. I still do not see the use of the hourly temporal resolution.

Further comments

Page 1 Line 6: "Increased temperature scenarios show that seasonal changes in snow-dynamics could offset a fairly constant negative change in relative runoff induced by evaporation, but not during the melt season."

I do not understand. What seasonal changes are you talking about? Where can I see this? What figure?

Page 2 Line 14: "higher snowmelt rates"

Lower snowmelt rates? https://www.nature.com/articles/nclimate3225

Page 2 Line 48: "high spatial and temporal resolutions ensure that small scale variability is accounted for."

I do not see how you account for small scale variability. The basic input data is very course (30x30 km) and only interpolated linearly to a 4 km grid. Strong spatial variability over short distances in the Alps, for example, is not captured.

Page 2 Line 54: "representative"

I do not get why it is 'representative'. Representative for other large river basin in Europe? Which ones? As you mention, it is a very heterogeneous basin, so this large heterogeneous basin actually is not representative for any sub-region in the basin, as they do not have this heterogeneity. Please check this again. Maybe I just don't get it. Page 2 Line 54: "north"

Northern Europe is roughly north of the southern coast of the Baltic Sea. It is more Central Europe, I think. Page 3 Line 63: "bilinear interpolation"

Why do yo use bilinear interpolation? Are there other ways to better address the spatial variability in the basin? Why do you interpolate on a 4x4 grid?

Page 3 Line 72: "yearly average precipitation sums decreasing from 1146 mm to 1066 mm"

How does this go along with mentioned 'intensification of the hydrological cycle'? Is this only decadal variability or a long-term trend? Can you compare to other 10-year slices?

Page 4 Figure 1: "climatic changes"

I am not sure weather it is appropriate here to call in climatic changes. You only compare to 10-year time slices only twenty years apart. Usually climatic changes are assessed over longer time periods comparing at least 30-year time slices. Particularly precipitation is subject to strong decadal variability. I don't think it is appropriate to attribute the differences in precipitation that you show to climate change.

Page 4 Line 92: "Given that dS2 is not calibrated on these variables, and the difference in spatial scale of the input data, this shows that dS2 is able to correctly simulate evaporation and snow processes."

This validation using only two point measurements does not really convince me. Are there other ways to better assess the performance in space? Maybe MODIS snow cover maps?

Page 6 Line 121: "confirming our hypothesis"

Your hypothesis was a linear response of discharge to temperature change?

Page 6 Line 132: "increased snowmelt"

Isn't it more liquid instead of solid precipitation?

Page 7 Figure 4: " Panel a shows the yearly average discharge under a 2.5° C "

Just for me to understand. First, you simulate the 1980s with your normal input data. Then you add 2.5°C on the temperature time series and re-run the model? So the increase in temperature strongly increases (linearly) the evaporation and hence discharge decreases? Does it makes sense that this effect is the same throughout the year?

Page 7 Line 133: "glaciers in the Alps"

There are glaciers in your model?

Page 8 Line 38: "As expected, the majority of the basin is controlled by the change induced by a change in evaporation (84–97%). As a result, the mean discharge is

reduced by ±18%."

Do you think that his is only in you model or is this effect the same in the 'real-world'? Any seasonal differences?

Page 8 Line 146: "interannual variability"

What about decadal variabilities?

Page 8 Line 147: "effect of different temperatures"

Also strong differences in precipitation between the two time slices!

Page 8 Line 149: "downscaling method"

A linear interpolation is better than downscaling? Isn't the simple bilinear interpolation you use adding a lot of errors and uncertainties throughout the basin?

Page 8 Line 159: "With higher temperatures, increased melt from glaciers and snow packs can offset the discharge reduction from enhanced evaporation over the majority of the year."

Where can I see this in your result figures? Please explain better what you mean by 'snow driven changes'.

Page 9 Line 168: "Enhanced melt will offset the negative trend caused by the increased evaporation, until the frozen water storages are depleted."

Where do you show this in your study?

―――――――――――――――――――――

---

## Referee Comment (RC2) · Anonymous Referee #2 · 29 Oct 2020

General comments: Buitink et al. showed the relative importance of changes in temperature, evaporation and precipitation on changes in discharges from the 1980s to the 2010s using the dS2 model in the Rhine river basin. The manuscript reads well and has little grammatical errors, but the structure and methods could use work to help readers understand the simulations. Information on the methodology is greatly missing, which causes readers to speculate how to interpret the overall conclusions. Also greatly missing are comparisons of this work to other modeling studies. Based on my review, I would suggest major revisions before publication is merited.

Major comments: 1. Section structure: The methods section is too brief. Much of the

model descriptions are contained in the supporting information and should be moved to the main text. The swapping of variables is confusing to me. How can you realistically change temperature only affecting snow processes or evaporation? Wouldn't both processes be affected by changing the temperature? If the goal of the paper is to simulate the hydrologic response to temperature-driven changes in evaporation and snow processes, specific details on the snow processes being simulated need to be included. The term 'snow processes' is used throughout the paper, but it is unclear which snow processes are simulated. Glacier melt is considered a snow process? Increased melt from glaciers is attributed to changes in discharge later in the manuscript. Is it possible to separate the effects from snowmelt and glacier melt?

The discussion section should be separated from the conclusion to compare your results with other studies and address the overall implications from your results better.

2. Figures: Figure 2 has a lot of results, but it is difficult to interpret due to the small size of the individual panels. Panels 2a, 2e, and 2f would benefit from being stretched out to see the results in more detail. Additional comments on Figure 2:

The colors in 2a do not match the legend. I assume the darker lines are for simulated and the lighter lines are observed, but clarification would be nice.

It appears that the sum of the difference in model simulations in 2e during February are cut off by the y-axis limits?

Why are there gaps in 2f? Does this mean that the forcing variables failed to explain any fraction?

It is confusing why the cumulative effects for forcing variables are shown in 2g, but the absolute effects are generally discussed in the text. This is how the 11%, 19%, and 18% from the abstract were calculated, correct?

All of the significant differences in monthly discharge shown in Fig. 2c are larger for the 1980s compared to the 2010s, yet the sum of the differences from the forcing variables

results in positive discharge differences during March and December in Fig. 2e. How is this possible?

The dotted grey line (sum of diff.) in Fig. 2e during early March is +500 m3/sec, but the solid grey line (2010s) shows a negative discharge difference for the same time, can you explain this? Similarly, during early May.

Occasionally text does not seem to align with results from the Figures:

Snow depth decreases for the majority Europe are reported in line 26 and shown in Figure 1e from ERA5 data, so why is the discharge difference from modified P in Figure 2d positive?

In lines 86-87 you write "Both variables are correctly represented, and show similar variability as the observations, even at hourly timescale." I would argue that snow storage is poorly simulated as the maximum simulated snow storage/height is twice as much as the observed maximum snow storage/height. This needs to be explained more. Could the positive discharge difference due to Modified P be due to the simulated maximum snow storage being twice as high as the observed maximum snow storage in Fig. 3b?

In lines 101-102 you write "During spring, this simulation shows higher discharge values resulting from increased snow melt." It is confusing which specific months you are referring to, but in Figs. 4b and 4c modified TSnow does not appear to have a positive effect on discharge. But during low flow conditions (Fig. 4d), TSnow has a positive effect on discharge?

In line 155: "Discharge between these two periods was significantly different for 8 out of 12 months". 10/12 boxes in 2b are full colored, representing significant differences.

Results are often grouped by months, but then seasonal changes are discussed in the text. This is confusing for the reader to speculate which particular months you are referring to. For instance, the third shaded period in Figure 4 is referred to in the text

as the "late summer low flow" period (line 120). But this time period aligns with the end of September through the beginning of November, not typically thought of as late summer. I would suggest referring to the results based on the monthly changes to remove this ambiguity.

3. Comparison between other studies: Much work has already been conducted on simulated effects of temperature changes on hydrologic response. It would be nice to see a comparison of your results to some of these studies and why your results agree or disagree from theirs. This appears to be completely missing. You list six studies that investigated these effects in lines 28-29, but do go on to discuss their results or compare yours at all.

Climate model simulations in western North America indicate that the fraction of melt-water volume produced at high snowmelt rates is greatly reduced in a warmer climate (i.e. "Slower snowmelt in a warmer world", Musselman et al., 2017). Additionally, model simulations suggest slower snowmelt decreases streamflow production ("Snowmelt rate dictates streamflow", Barnhart et al., 2016). But, in lines 161 – 163 you write "With higher temperatures, increased melt from glaciers and snow packs can offset the discharge reduction from enhanced evaporation over the majority of the year" and that "these results can be interpreted for the many different basins around the globe depending on both rain- and snowfall". These sentences are confusing and incredibly misleading. Your remarks make it seem like increased snowmelt and glacier melt off-sets reduction from evaporation and results in an inconsequential effect on discharge. But in Figures 2b and 2c you show significantly lower discharge for 10/12 months and annually for the 2010s compared to the 1980s.

Minor corrections: Line 14: Higher temperatures have been shown to lead to slower snowmelt rates ("Slower snowmelt in a warmer world", Musselman et al. 2017, Nature Climate).

Lines 18-21: Changes in discharge are most likely to be strongly affected by changes

in precipitation. It's probably best to focus on the runoff ratio (discharge/precipitation). Runoff ratios (or runoff efficiency) were found to be mostly unchanged in snow-covered areas of the western U.S. despite increasing temperatures and decreased snow fractions ("Warming is Driving Decreases in Snow Fractions While Runoff Efficiency Remains Mostly Unchanged in Snow-Covered Areas of the Western United States"; McCabe et al., 2018; Hydrometeorology).

Line 23: "Europe has experienced significant changes in evaporation, snow depth and streamflow over the last decades". Citation needed. Were all of the changes negative? Which decades?

Lines 24-25: "Their study shows that both changes in precipitation and evaporation had considerable effects on the streamflow." Did they observe negative changes for both precipitation and evaporation? How do changes in rain compare to changes in snow?

Line 26: "showed that snow depth decreased over the majority of Europe". From when to when?

Lines 37-38: "for example, the study by Mastrotheodoros et al. (2020) took more than $6 \times 105$ CPU hours". What resolution did they use?

Lines 42 - 44: "This study investigates the hydrological response to temperature-driven changes in evaporation and snow processes, testing our main hypotheses that both seasonal changes in snowmelt and enhanced evaporation will aggravate low flows, and that the changes will increase with temperature under realistic warming." Is snowmelt the only snow process tested in this study? If so, I would change the terms "snow processes" to "snow melt".

Lines 53-54: "The Rhine basin was selected because the climate and basin heterogeneity are representative for north-western Europe and many other basins globally". It seems like a stretch to suggest the Rhine basin is representative of many basins globally.

Line 80: "higher flows during late winter in the 2010s". Be specific about the months. It seems that from 2b discharge is lower for all months in the 2010s. I do not see higher flows during late winter in the 2010s?

Lines 101-103: "The higher temperatures of the 2010s also resulted in lower discharge values in the first few months of the year. During spring, this simulation shows higher discharge values resulting from increased snow melt." Again, it is confusing which months you are referring to. I would assume that the first few months of the year are late winter. These two sentences seem contradictory. First you say that higher temperature (affecting snowmelt) resulted in lower discharge, then you say higher temperature (affecting snowmelt) resulted in higher discharge. Was the increase in spring discharge due to an increase in the snowmelt rate of the volume of snowmelt?

Lines 107-108: "The explained fraction is lowest during spring and late summer." Be specific about which months. Are you referring to the gaps in Fig. 2f?

Line 120: "late summer low flow". Be specific about which months. The late summer period aligns with the end of September through early November, more typical of fall/autumn. Figure 4b: The effects from modified TEvap in Figure 4b are hidden by the combined effects.

Line 155: "Discharge between these two periods was significantly different for 8 out of 12 months." Is this simulated or observed? In Fig. 2c you show all but two months being significantly different?

Lines 162-163: "With higher temperatures, increased melt from glaciers and snow packs can offset the discharge reduction from enhanced evaporation over the majority of the year." This interpretation is opposite of previously published studies (i.e. slower snowmelt in a warmer world) and needs clarified.

---

## Referee Comment (RC3) · Anonymous Referee #3 · 7 Nov 2020

The paper "Seasonal discharge response to temperature-driven changes in evaporation and snow processes" aims at testing the hypotheses that both seasonal changes in snowmelt and enhanced evaporation can exacerbate low flows, and that changes will increase with temperature under realistic warming. I think this study fits very well with the scope of ESD journal and it addresses an important and timely topic. However, the authors should first address some major issues before this manuscript can be considered for possible publication.

My first concern is related to the structure and readability of the paper. I found the description of the method really poor, fragmented, and the results section is a mix of both

findings and methods. Overall, I believe that this paper structure makes the manuscript confusing and difficult to read. Why not add a "case study" and "Experimental setup" sections before describing the results? Also, the authors described figure 2.a and then they jumped to figure 3, while the remaining part of figure 2 is described only after. A solution could be to split figure 2 into different ones. Try to be more consistent.

My second concern regards the swapped method introduced in this study to understand how the individual forcing variables affect the hydrological cycle. I found this approach quite atypical, and an adequate justification should be included for why such an approach is employed. I am not deeply familiar with this swapped approach, but there is no reasoning provided for why such an approach should be preferred over other statistical approaches.

Another serious issue is the model structure. Several major hydraulic works and flood control measures were constructed over the years in the Rhine basin, strongly modifying its hydrological cycle and flood responses. How are those structures included in your distributed efficient hydrological model? This can be a major issue as major hydraulic works may have a higher influence than the forcing variables analyzed in this paper, thus compromising the findings and conclusions of the study.

Linking to the structural issue of the manuscript, the authors can clearly mention in the new "Experimental setup" section that the swapped and changing-temperature approaches are meant to answer the two hypotheses of this study (see introduction). Moreover, to further improve the readability of the paper, the authors could better connect the two hypotheses of the introduction with the results summarized in the conclusions. Right now everything is there, but it takes quite some time to grasp the main take-home message.

The authors mentioned in the conclusions that "Here we selected a resolution of $4 \times 4$ km, so we can use the ERA5 forcing data without downscaling methods (adding uncertainty and potential errors)". However, if ERA5 has a resolution of 0.25degree, how

it is possible not to downscale the dataset to adapt it to a higher resolution of 4km?

". . . yet these results can be interpreted for the many different basins around the globe depending on both rain- and snowfall". This sentence should be rephrased as you do not know what can occur over other basins with totally different characteristics. The results of this study cannot be generalized to other studies without proper large-scale validations. The same applies to other generalizations introduced in the conclusions. Include study limitations in the conclusion section

---

## Author Comment (AC1) · 19 Nov 2020

We would like to thank Anonymous Referee #1 (AR1) to write this review. Below, we will reply to the points made by AR1, with the comments from AR1 in black, and our response in blue.

**General comment**

5 Buitink et al. simulate the Rhine River Basin for two 10-year time slices (1980s and 2000s) using the dS2 model and ERA5 data (0.25x0.25). An additional soil moisture model was added in order to attain values of actual evapotranspiration. ERA5 data was interpolated to a 4x4 km grid using bilinear interpolation. They assess changes in discharge between the two time slices and attribute differences in runoff to differences in precipitation, evapotranspiration and snowmelt. In general, it is an interesting study and has the potential to become a valuable contribution to hydrological research. However, I see several major

10 issues regarding the analysis and the text that need to be addressed before it can be considered for publication.
    Thanks for your overall positive evaluation of our manuscript, and the suggestions for improvement.

**Major comments**

**Comment 1: Structure text**

One major issue I see is that sections in the manuscript are mixed up or even missing. Often, model and method description

15 are located in the result section. The results section is mixed with the discussion. In general, the manuscript lacks important details. I think a more detailed method description, where each step of the analysis conducted is described, needs to be added. Furthermore, model set-up and model components need to be described better. This is crucial to understand you experiments. Please provide more information on the dS2 model and the snow routine. An additional section 'study area and data' could be good to better introduce the Rhine Basin. In my opinion, large parts of the supplementary material can be moved into the

20 actual manuscript.
    We understand the confusion caused by the structure of the text, as this comment is also discussed by AR2 and AR3. It was an effort to write a short and concise manuscript, but we now see that we need to provide more explanation to improve readability of the manuscript. We propose to move the model description from the supplement to the main text. We will also add information to describe and introduce the Rhine basin. Additionally, we will add headers to clearly separate the different

25 experiments performed in this study. We will extend the methods description to improve the explanation of the forcing swap variables (see below).

**Comment 2: Swapping variables**

I still have troubles to understand your approach of swapping forcing variables in order to attribute changes. You try to asses the contribution of the three factors: 1) changes in precipitation, 2) changes in snowmelt due to changes in temperature and 3)

30 changes in evapotranspiration due to changes in temperature. If I understand correctly, you swap individual forcing variables (temperature or precipitation, respectively) between the two 10-year time slices. You run the 1980s with temperature data from the 2000s, for example. So the temperature in the first week of August 2005 becomes the temperature in the first week of 1985, right?
    This is correct. We agree in hindsight that a proper description of this experiment is lacking, hence we will add a paragraph

35 in the methods section to explain this experiment. The original conceptual figure of 2d is not enough to properly describe the workflow. Therefore, we propose to add a new conceptual figure together with explanations and equations (added at the end of this reply). We hope this clarifies the experiment.
    This seems to be a very rough approach and I am not sure if this is a good idea. Also the performance of this approach seems quite poor. For parts of the year you can not at all explain variations or even expect an opposite trend (in February, for

40 example, strong overestimation and in March even an opposite trend). The three factors you investigate do not really explain the variations in discharge, I think.

Figure 2a shows that the full model is able to correctly simulate the discharge in both periods (1980s and 2010s). As this is the same calibrated model, the only changes between these simulations are in the forcing data (precipitation and temperatures). The three factors (forcing) that are varied are the only things that change, so any change unexplained by the sum of the

45    individual factors is not poor model performance, but reflects complexity and non-linearity in the overall response. We feel that the use of "unexplained" was probably falsely suggestive of poor model performance, so the terminology will be changed in the revision.

All variations that you can not explain, you attribute to 'interactions'. To me, these 'interactions' are not clear. I do not know what you have in mind here. Can you explain more detailed?

50    With "interactions" we try to describe the (nonlinear) interplay between the forcing variables. For example: precipitation can either fall as snow or rainfall, depending on the temperature. A change in temperature will therefore also influence the precipitation dynamics.

It looks like that a model only using the factor 'changes in precipitation' alone would explain changes in discharge better than the approach with the three factors (Fig. 2e). Please check.

55    Precipitation does seem indeed to have a very large influence, which is not unexpected given the precipitation differences between the two periods.

What about changes in evapotranspiration due to changes in precipitation? What about changes in snowmelt due to changes in precipitation?

These are also present, and this is a nice example of what we refer to as "interactions". We hope that the new method section

60    (see below) will clarify this.

**Comment 3: Attribution**

In the abstract you write that variation can be 'explained by the changes induced by snow (11%), evaporation (19%) and precipitation (18%), while 52% was driven by combination of these variables." This bases on results presented in Fig. 2 panels e, f and g, I assume. How do you calculate those percentage values? In February and April, for example, you have a negative

65    variation in discharge between the two 10-year slices of around 1000 m3 /s. This variations can be explained by variation in rainfall to 90-100%, right?

These percentages are indeed based on the results in Fig. 2e–g. As mentioned earlier, we understand the confusion and hope that a better explanation will clarify our workflow. Rainfall can indeed explain large fractions of the change, but we also need to take the changes induced by temperature (both evaporation and snow) into account, as these also change during between the

70    two periods. This is the reason why we did not calculate the total contribution of only precipitation changes, but always take these changes with reference to the combined changes (P + Tevap + Tsnow).

**Comment 4: Evapotranspiration**

Are you using evaporation or evaporation + transpiration = evapotranspiration? You mention the Penman-Monteith equation to get potential evaporation. Isn't it reference evapotranspiration you get? Do you calculate this? Or is it ERA5 data directly?

75    You show a map of the potential evaporation (Fig. 1c). Can you also show a map of the calculated actual evaporation? Please give more information on the soil hydraulic data used. If I understand correctly, you assume one value of the rootzone depth for the entire basin? Is this value also constant over time? What about vegetation cover?

We do indeed calculate reference evapotranspiration, based on ERA5 data. We will add a map to validate the calculated evapotranspiration by comparing it to GLEAM data. We will move information on the model and data from the supplement to

80    the main manuscript. We do indeed assume one value of rootzone over the entire basin, constant in both space and time.

**Comment 5: Temperature 'scenarios'**

In my opinion, the different approaches you use need to be explained better. Your swapping approach and the temperature 'scenarios'. Please explain in the method section. In those temperature 'scenarios', you simple add e.g. 2 ◦C to the hourly temperature time series? This again seems a very simple approach. In what way do those increases influence your model

85    components and does this reflect 'real-world' processes? In Fig. 5 b your results indicate that increasing temperatures will

rise evaporation in winter, in turn, decreasing discharges. According to my knowledge, evapotranspiration does barely play any role in winter (low radiation, low temperatures, no plants). All studies I know hint at an increase in discharge in winter, e.g. due to liquid instead of solid precipitation. Can you check this and maybe show your simulated evapotranspiration values for winter for this temperature 'scenarios'? Your model shows a linear response of evapotranspiration to temperature. How is evapotranspiration affected by changes in temperatures in the first place? Isn't it an input to your model?

We will clarify the different experiments in the next version of the manuscript. The discharge reduction visible in Fig. 5b can indeed be attributed to increased evaporation, but also a change in water storage. With a change in evaporation (which has indeed the largest influence during summer), water stored in the catchment will be reduced. This reduction continues to affect discharges during winter, as less storage means less discharge production. The change from solid to liquid evaporation is not included in the Tevap run, as this is part of the Tsnow simulation, where we do indeed see an increase in discharge during the months November–May.

These results does seem to show a rather linear response to temperature, although there appears to be a slight curvature in some responses (Tsnow in Fig. 4b, Tevap and Tboth in Fig. 4d). As these processes are threshold processes (e.g. snow fall and melt), you would expect a non-linear response. We expect that this becomes more visible when reaching more extreme ends of the hydrological spectrum (droughts and/or floods), but this is not clearly visible from our analysis.

**Comment 6: Validation, Calibration**

Why do you use two point measurements of this small pre-Alpine catchment to validate? Isn't there hundreds of snow gauges? Where is this snow gauge exactly? At what elevation? You compare the simulated snow in your 4x4km cell with the point measurement? Why do you simulate in an hourly resolution? Why not daily? For the routing? Can't you use daily data for calibration and validation (even if you initially run hourly)? At the end you aggregate to monthly values anyway. I still do not see the use of the hourly temporal resolution.

We choose to use an hourly resolution to ensure a correct link between space and time, as the choice of hourly resolution can have a large effect on model output (Melsen et al., 2016). For example, an hourly resolution can capture the snow melt dynamics during a single day, which is impossible at the daily timestep.

We validate the model with these point observations, as they are also hourly observations. With this validation, we validate our model in time, on variables which we did not optimize. This proves to give more confidence that the model is able to correctly simulate the fluxes important for our study. As far as we know, most snow observations are daily records of snow depth rather than continuous records at hourly resolution. In the revision we will provide more details about the experimental Rietholzbach catchment were the observations were made. We will add a spatial validation as well, by comparing the snow cover with observed snowcover, and by comparing our evapotranspiration with GLEAM evapotranspiration.

**Further comments**

Page 1 Line 6: "Increased temperature scenarios show that seasonal changes in snowdynamics could offset a fairly constant negative change in relative runoff induced by evaporation, but not during the melt season." I do not understand. What seasonal changes are you talking about? Where can I see this? What figure?

This is visible in Fig. 4, where the purple line is above the 1980s simulatino for the majority of the year. We will rephrase this sentence.

Page 2 Line 14: "higher snowmelt rates" Lower snowmelt rates? https://www.nature.com/articles/nclimate3225

Thanks for this reference. We will discuss this work in a new discussion section to place our study in existing work.

Page 2 Line 48: "high spatial and temporal resolutions ensure that small scale variability is accounted for." I do not see how you account for small scale variability. The basic input data is very course (30x30 km) and only interpolated linearly to a 4 km grid. Strong spatial variability over short distances in the Alps, for example, is not captured.

Here we refer to temporal variability (e.g. snow melt patterns throughout the day). We will rephrase this.

Page 2 Line 54: "representative" I do not get why it is 'representative'. Representative for other large river basin in Europe? Which ones? As you mention, it is a very heterogeneous basin, so this large heterogeneous basin actually is not representative for any sub-region in the basin, as they do not have this heterogeneity. Please check this again. Maybe I just don't get it.

This is indeed misleading, and we will rephrase this.

Page 2 Line 54: "north" Northern Europe is roughly north of the southern coast of the Baltic Sea. It is more Central Europe, I think.

We will rephrase this.

Page 3 Line 63: "bilinear interpolation" Why do yo use bilinear interpolation? Are there other ways to better address the spatial variability in the basin? Why do you interpolate on a 4x4 grid?

We chose to use a 4x4 km grid to strike a balance between input data, spatial variability and run times. We decided to interpolate using bilinear interpolation to avoid adding additional uncertainty. We are aware that this method also adds uncertainty, but so does running a hydrological model at 0.25x0.25 degree resolution.

Page 3 Line 72: "yearly average precipitation sums decreasing from 1146 mm to 1066 mm" How does this go along with mentioned 'intensification of the hydrological cycle'? Is this only decadal variability or a long-term trend? Can you compare to other 10-year slices?

This can indeed be attributed to decadal variability. As the focus of our manuscript is not on precipitation changes, we decided not to go further into the cause and effect of precipitation changes.

Page 4 Figure 1: "climatic changes" I am not sure weather it is appropriate here to call in climatic changes. You only compare to 10-year time slices only twenty years apart. Usually climatic changes are assessed over longer time periods comparing at least 30-year time slices. Particularly precipitation is subject to strong decadal variability. I don't think it is appropriate to attribute the differences in precipitation that you show to climate change.

With climatic we refer to hydro-meteorological changes. We will rephrase this.

Page 4 Line 92: "Given that dS2 is not calibrated on these variables, and the difference in spatial scale of the input data, this shows that dS2 is able to correctly simulate evaporation and snow processes." This validation using only two point measurements does not really convince me. Are there other ways to better assess the performance in space? Maybe MODIS snow cover maps?

We see this a temporal validation, but we will extend our validation to a spatial validation on observed snow cover and evaporation from GLEAM.

Page 6 Line 121: "confirming our hypothesis" Your hypothesis was a linear response of discharge to temperature change?

The sign of the change was inline with our hypothesis, we will clarify this.

Page 6 Line 132: "increased snowmelt" Isn't it more liquid instead of solid precipitation?

A combination of more melt and liquid precipitation. We will change this.

Page 7 Figure 4: " Panel a shows the yearly average discharge under a 2.5∘ C " Just for me to understand. First, you simulate the 1980s with your normal input data. Then you add 2.5∘C on the temperature time series and re-run the model? So the increase in temperature strongly increases (linearly) the evaporation and hence discharge decreases? Does it makes sense that this effect is the same throughout the year?

This description is correct. See our reply below Comment 5 for an explanation of this behaviour.

Page 7 Line 133: "glaciers in the Alps" There are glaciers in your model?

Glaciers are represented in dS2, although they do not have a separate module, but rather are pixels with excessive amounts of snow.

Page 8 Line 38: "As expected, the majority of the basin is controlled by the change induced by a change in evaporation (84–97%). As a result, the mean discharge is reduced by ±18%." Do you think that his is only in you model or is this effect the same in the 'real-world'? Any seasonal differences?

There are indeed seasonal differences. The results presented here are annual values, and are not the same for every season. We expect this also to be realistic, as we already see a decrease in discharge between the 1980s and the 2010s (partly due to changes in precipitation as well).

Page 8 Line 146: "interannual variability" What about decadal variabilities?

As we are mostly focussing on the changes in temperatures, it is known that the difference between the periods can be attributed to changes in climate.

Page 8 Line 147: "effect of different temperatures" Also strong differences in precipitation between the two time slices!

This is true, but we mainly focus on the changes in temperature.

Page 8 Line 149: "downscaling method" A linear interpolation is better than downscaling? Isn't the simple bilinear interpolation you use adding a lot of errors and uncertainties throughout the basin?

We don't argue that linear interpolation is the better method, just that it has less degrees of freedom.

Page 8 Line 159: "With higher temperatures, increased melt from glaciers and snow packs can offset the discharge reduction from enhanced evaporation over the majority of the year." Where can I see this in your result figures? Please explain better what you mean by 'snow driven changes'.

We understand the confusion around the "snow driven changes" and "snow processes" terms. We use this to group all snow and ice related processes (snow fall, melt, etc), and we will make sure this is properly explained in the next version of the manuscript. This "offset" is visible in Fig. 4a, where the Tsnow line shows higher discharge values than the 1980s simulation for the majority of the year.

Page 9 Line 168: "Enhanced melt will offset the negative trend caused by the increased evaporation, until the frozen water storages are depleted." Where do you show this in your study?

This is visible in Fig. 4a, where the Tsnow line shows higher discharge values than the 1980s simulation for the majority of the year. We understand that we do not directly show the frozen water storages, and will clarify this.

**1 Methods - Forcing swap**

In the first experiment, we aim to understand how each forcing variable can explain the resulting changes in discharge, and their relative importance. To perform this, we setup the experiment according to the conceptual overview presented in Fig. 1. In order to investigate how temperature influences evapotranspiration and snow processes separately, we perform model runs in which the total temperature change is splitted into temperature effects on evapotranspiration ("Changed $T_{evap}$") and snow processes ("Changed $T_{snow}$"). In addition, another run is performed with only changes in P ("Changed P"), so that these individual runs can be compared to a run where all changes in forcing are enabled (2010s forcing). The resulting simulated discharge is compared to the 1980s run, to determine the discharge change. In this way, we can evaluate the relative impact of each forcing variable on the discharge.

We sum the discharge changes of the three forcing swapped runs, to obtain $\mathrm{Sum}\Delta$:

$$\mathrm{Sum}\Delta = \Delta Q_P + \Delta Q_{T\ snow} + \Delta Q_{T\ evap} \tag{1}$$

where $\Delta Q_x$ represents the discharge difference of the forcing swapped simulations. We can use this $\mathrm{Sum}\Delta$ to study how well it explains the 2010s run, by comparing it to $\Delta Q_{2010s}$. We hypothesise that when $\mathrm{Sum}\Delta$ is equal to $\Delta Q_{2010s}$, the effect of the forcing is additive, and together explain all differents. We will refer to this as the direct effects. In the case of a discrepancy between $\mathrm{Sum}\Delta$ and $\Delta Q_{2010s}$, this can be attributed to interaction between the three forcing components. We will refer to this as indirect effects. We define $\mathrm{Sum}\Delta$ to have explanatory value when it has the same sign as $\Delta Q_{2010s}$. We calculate the contribution of the direct effects ($\phi$) using the following equation:

$$\phi = \begin{cases} \frac{\min(\mathrm{Sum}\Delta, \Delta Q_{all})}{\max(\mathrm{Sum}\Delta, \Delta Q_{all})}, & \text{if } \mathrm{sign}(\mathrm{Sum}\Delta) = \mathrm{sign}(\Delta Q_{all}) \\ 0, & \text{if } \mathrm{sign}(\mathrm{Sum}\Delta) \neq \mathrm{sign}(\Delta Q_{all}) \end{cases} \tag{2}$$

This value can then be used to calculate the relative (direct) contribution of each forcing variable, using the following equation:

$$\phi_x = \frac{\mathrm{abs}(\Delta Q_x)}{(\mathrm{abs}(\Delta Q_P) + \mathrm{abs}(\Delta Q_{T\ snow}) + \mathrm{abs}(\Delta Q_{T\ evap}))} \cdot \phi \tag{3}$$

where $\Delta Q_x$ should be replaced by $\Delta Q_P$, $\Delta Q_{T\ snow}$, or $\Delta Q_{T\ evap}$.

[Figure]

**Figure 1.** Conceptualization of the forcing swap experiment, showing the different simulations (rows) and steps in the analysis (columns). The different forcing variables are visualized as colored blocks, where the solid and dashed boxes indicate forcing data from the 1980s and the 2010s, respectively.

**References**

215    Melsen, L. A., Teuling, A. J., Torfs, P. J. J. F., Uijlenhoet, R., Mizukami, N., and Clark, M. P.: HESS Opinions: The need for process-based evaluation of large-domain hyper-resolution models, Hydrology and Earth System Sciences, 20, 1069–1079, https://doi.org/10.5194/hess-20-1069-2016, http://www.hydrol-earth-syst-sci.net/20/1069/2016/, 2016.

---

## Author Comment (AC2) · 19 Nov 2020

We would like to thank Anonymous Referee #2 (AR2) to write this review. Below, we will reply to the points made by AR2, with the comments from AR2 in black, and our response in blue.

**General comments:**

Buitink et al. showed the relative importance of changes in temperature, evaporation and precipitation on changes in discharges from the 1980s to the 2010s using the dS2 model in the Rhine river basin. The manuscript reads well and has little grammatical errors, but the structure and methods could use work to help readers understand the simulations. Information on the methodology is greatly missing, which causes readers to speculate how to interpret the overall conclusions. Also greatly missing are comparisons of this work to other modeling studies. Based on my review, I would suggest major revisions before publication is merited.

Thanks for writing this review. We agree with the comments made, and believe this will improve the quality of the manuscript.

**Major comments:**

**1. Section structure:**

The methods section is too brief. Much of the model descriptions are contained in the supporting information and should be moved to the main text. The swapping of variables is confusing to me. How can you realistically change temperature only affecting snow processes or evaporation? Wouldn't both processes be affected by changing the temperature? If the goal of the paper is to simulate the hydrologic response to temperature-driven changes in evaporation and snow processes, specific details on the snow processes being simulated need to be included. The term 'snow processes' is used throughout the paper, but it is unclear which snow processes are simulated. Glacier melt is considered a snow process? Increased melt from glaciers is attributed to changes in discharge later in the manuscript. Is it possible to separate the effects from snowmelt and glacier melt? The discussion section should be separated from the conclusion to compare your results with other studies and address the overall implications from your results better.

This point was also mentioned by the other reviewers. We will move the model descriptions from the supplement to the main manuscript, and add a description on the forcing swapping (see below). We understand the confusion regarding our usage of the term "snow processes", as it is indeed a term grouping snowfall, and melt from snow and ice: we will clarify this in the next version. We will try to separate the effects cause by snowmelt and glaciermelt, as this is already partially visible in Fig. 5a (blue pixels during the second and third periods). We agree that a comparison with other studies is currently lacking, and we will add this in the next version of the manuscript.

**2. Figures:**

Figure 2 has a lot of results, but it is difficult to interpret due to the small size of the individual panels. Panels 2a, 2e, and 2f would benefit from being stretched out to see the results in more detail.

We agree that Fig. 2 contains a lot of information. We have decided to split the figure into 2 figures, and replace Fig. 2d with an improved version to visualise the method behind forcing-swapping (see below)

Additional comments on Figure 2:

– The colors in 2a do not match the legend. I assume the darker lines are for simulated and the lighter lines are observed, but clarification would be nice.

This is correct, and we will clarify this in the next version of the manuscript.

– It appears that the sum of the difference in model simulations in 2e during February are cut off by the y-axis limits?

This is correct and we will fix this.

40     – Why are there gaps in 2f? Does this mean that the forcing variables failed to explain any fraction?

These are no gaps, but show that the forcing variables failed to explain the difference. We understand the confusion, and hope that our new description and figure on this method (see below) clarifies the results in Fig. 2f.

– It is confusing why the cumulative effects for forcing variables are shown in 2g, but the absolute effects are generally discussed in the text. This is how the 11%, 19%, and 18% from the abstract were calculated, correct?

45     – All of the significant differences in monthly discharge shown in Fig. 2c are larger for the 1980s compared to the 2010s, yet the sum of the differences from the forcing variables results in positive discharge differences during March and December in Fig. 2e. How is this possible?

This is caused by the fact that the sum of the differences failed to explain the real 2010s discharge (see also the low fraction explained during March). This can again be related to the "forcing interactions"

50     – The dotted grey line (sum of diff.) in Fig. 2e during early March is +500 m3/sec, but the solid grey line (2010s) shows a negative discharge difference for the same time, can you explain this? Similarly, during early May.

Again the result of interactions between the forcing variables. For example, during March, the sum of differences expects higher discharge values, while 2010s was indeed lower. As the forcing swapped runs failed to capture the interaction between precipitation and temperature (snowfall and melt, evaporation and other storage related processes), it also failed
55     to get close to the 2010s values.

Occasionally text does not seem to align with results from the Figures:

– Snow depth decreases for the majority Europe are reported in line 26 and shown in Figure 1e from ERA5 data, so why is the discharge difference from modified P in Figure 2d positive?

The data in Fig. 2d is synthetic and not based on real results, at it was an effort to conceptualize the calculations performed
60     in Fig. 2e and f. As mentioned earlier, we will remove this figure with an improved conceptual figure.

– In lines 86-87 you write "Both variables are correctly represented, and show similar variability as the observations, even at hourly timescale." I would argue that snow storage is poorly simulated as the maximum simulated snow storage/height is twice as much as the observed maximum snow storage/height. This needs to be explained more. Could the positive discharge difference due to Modified P be due to the simulated maximum snow storage being twice as high as the
65     observed maximum snow storage in Fig. 3b?

There is a difference between the plotted variables: the colored line (simulation) shows the snow storage in snow water equivalent (SWE), where the black line shows the observations as snow height. A direct translation from snow height to SWE is difficult due to snow compaction. This is why we only focus on dynamics.

– In lines 101-102 you write "During spring, this simulation shows higher discharge values resulting from increased snow
70     melt." It is confusing which specific months you are referring to, but in Figs. 4b and 4c modified TSnow does not appear to have a positive effect on discharge. But during low flow conditions (Fig. 4d), TSnow has a positive effect on discharge?

In lines 101-102 we refer to the months March–May, were the simulation with modified Tsnow shows higher discharge values (as visible in Fig. 2e). In this experiment, we replace the temperature time series of 1980s with the time series from 2010s. In Fig. 4 we simply increase the temperature of the 1980s time series. In the three sub-panels, we do not
75     focus on the period March–May, but on the grey highlighted regions in Fig. 4a. Despite this, the increased discharge during March–May is still visible in Fig. 4a. In the next version of the manuscript, we will make sure that we better clarify the periods we are referring to.

– In line 155: "Discharge between these two periods was significantly different for 8 out of 12 months". 10/12 boxes in 2b are full colored, representing significant differences.

    The text is wrong, this should indeed be 10 out of 12 months.

– Results are often grouped by months, but then seasonal changes are discussed in the text. This is confusing for the reader to speculate which particular months you are referring to. For instance, the third shaded period in Figure 4 is referred to in the text as the "late summer low flow" period (line 120). But this time period aligns with the end of September through the beginning of November, not typically thought of as late summer. I would suggest referring to the results based on the monthly changes to remove this ambiguity.

Thanks for this point, and we will clarify this in the next version of the manuscript.

**3. Comparison between other studies**

Much work has already been conducted on simulated effects of temperature changes on hydrologic response. It would be nice to see a comparison of your results to some of these studies and why your results agree or disagree from theirs. This appears to be completely missing. You list six studies that investigated these effects in lines 28-29, but do go on to discuss their results or compare yours at all.

This is a valid argument: we will add a section to compare our results to similar studies.

Climate model simulations in western North America indicate that the fraction of meltwater volume produced at high snowmelt rates is greatly reduced in a warmer climate (i.e. "Slower snowmelt in a warmer world", Musselman et al., 2017). Additionally, model simulations suggest slower snowmelt decreases streamflow production ("Snowmelt rate dictates streamflow", Barnhart et al., 2016). But, in lines 161 – 163 you write "With higher temperatures, increased melt from glaciers and snow packs can offset the discharge reduction from enhanced evaporation over the majority of the year" and that "these results can be interpreted for the many different basins around the globe depending on both rain- and snowfall". These sentences are confusing and incredibly misleading. Your remarks make it seem like increased snowmelt and glacier melt offsets reduction from evaporation and results in an inconsequential effect on discharge. But in Figures 2b and 2c you show significantly lower discharge for 10/12 months and annually for the 2010s compared to the 1980s.

We agree that these sentences can be misleading, and will rephrase those. With "offset" we mean that the change resulting from Tsnow make the discharge less severe, but not that it can fully compensate for the increased evaporation.

**Minor corrections:**

Line 14: Higher temperatures have been shown to lead to slower snowmelt rates ("Slower snowmelt in a warmer world", Musselman et al. 2017, Nature Climate).

We will discuss this in a section comparing our work with other studies.

Lines 18-21: Changes in discharge are most likely to be strongly affected by changes in precipitation. It's probably best to focus on the runoff ratio (discharge/precipitation). Runoff ratios (or runoff efficiency) were found to be mostly unchanged in snow-covered areas of the western U.S. despite increasing temperatures and decreased snow fractions ("Warming is Driving Decreases in Snow Fractions While Runoff Efficiency Remains Mostly Unchanged in Snow-Covered Areas of the Western United States"; McCabe et al., 2018; Hydrometeorology).

We will discuss this in a section comparing our work with other studies.

Line 23: "Europe has experienced significant changes in evaporation, snow depth and streamflow over the last decades". Citation needed. Were all of the changes negative? Which decades?

We refer to the relevant studies in the next studies of this paragraph.

Lines 24-25: "Their study shows that both changes in precipitation and evaporation had considerable effects on the streamflow." Did they observe negative changes for both precipitation and evaporation? How do changes in rain compare to changes in snow?

Line 26: "showed that snow depth decreased over the majority of Europe". From when to when?

They showed this to be a trend since the 1950s.

Lines 37-38: "for example, the study by Mastrotheodoros et al. (2020) took more than $6 \times 105$ CPU hours". What resolution did they use?

A higher resolution of 250x250 m. This will be mentioned to put the difference in a correct perspective, but the difference
125   in resolution alone cannot explain the much larger difference in computer time.

Lines 42 - 44: "This study investigates the hydrological response to temperature-driven changes in evaporation and snow processes, testing our main hypotheses that both seasonal changes in snowmelt and enhanced evaporation will aggravate low flows, and that the changes will increase with temperature under realistic warming." Is snowmelt the only snow process tested in this study? If so, I would change the terms "snow processes" to "snow melt".
130   Also changes in snowfall. We understand the confusion, and will define "snow processes" in the next version.

Lines 53-54: "The Rhine basin was selected because the climate and basin heterogeneity are representative for north-western Europe and many other basins globally". It seems like a stretch to suggest the Rhine basin is representative of many basins globally

We will rephrase this.
135   Line 80: "higher flows during late winter in the 2010s". Be specific about the months. It seems that from 2b discharge is lower for all months in the 2010s. I do not see higher flows during late winter in the 2010s?

This is indeed incorrectly written, we will correct this.

Lines 101-103: "The higher temperatures of the 2010s also resulted in lower discharge values in the first few months of the year. During spring, this simulation shows higher discharge values resulting from increased snow melt." Again, it is confusing
140   which months you are referring to. I would assume that the first few months of the year are late winter. These two sentences seem contradictory. First you say that higher temperature (affecting snowmelt) resulted in lower discharge, then you say higher temperature (affecting snowmelt) resulted in higher discharge. Was the increase in spring discharge due to an increase in the snowmelt rate of the volume of snowmelt?

We understand the confusion, and will clarify this, including specific references to the months.
145   Lines 107-108: "The explained fraction is lowest during spring and late summer." Be specific about which months. Are you referring to the gaps in Fig. 2f?

Yes, we are referring to the gaps, we will clarify this.

Line 120: "late summer low flow". Be specific about which months. The late summer period aligns with the end of September through early November, more typical of fall/autumn. Figure 4b: The effects from modified TEvap in Figure 4b are hidden by
150   the combined effects.

We will clarify this.

Line 155: "Discharge between these two periods was significantly different for 8 out of 12 months." Is this simulated or observed? In Fig. 2c you show all but two months being significantly different?

Simulated, and the 8/12 should indeed be 10/12.
155   Lines 162-163: "With higher temperatures, increased melt from glaciers and snow packs can offset the discharge reduction from enhanced evaporation over the majority of the year." This interpretation is opposite of previously published studies (i.e. slower snowmelt in a warmer world) and needs clarified.

We will clarify this in the next version.

**1 Methods - Forcing swap**

In the first experiment, we aim to understand how each forcing variable can explain the resulting changes in discharge, and their relative importance. To perform this, we setup the experiment according to the conceptual overview presented in Fig. 1. In order to investigate how temperature influences evapotranspiration and snow processes separately, we perform model runs in which the total temperature change is splitted into temperature effects on evapotranspiration ("Changed $T_{evap}$") and snow processes ("Changed $T_{snow}$"). In addition, another run is performed with only changes in P ("Changed P"), so that these individual runs can be compared to a run where all changes in forcing are enabled (2010s forcing). The resulting simulated discharge is compared to the 1980s run, to determine the discharge change. In this way, we can evaluate the relative impact of each forcing variable on the discharge.

We sum the discharge changes of the three forcing swapped runs, to obtain $\mathrm{Sum}\Delta$:

$$\mathrm{Sum}\Delta = \Delta Q_P + \Delta Q_{T\ snow} + \Delta Q_{T\ evap} \tag{1}$$

where $\Delta Q_x$ represents the discharge difference of the forcing swapped simulations. We can use this $\mathrm{Sum}\Delta$ to study how well it explains the 2010s run, by comparing it to $\Delta Q_{2010s}$. We hypothesise that when $\mathrm{Sum}\Delta$ is equal to $\Delta Q_{2010s}$, the effect of the forcing is additive, and together explain all differents. We will refer to this as the direct effects. In the case of a discrepancy between $\mathrm{Sum}\Delta$ and $\Delta Q_{2010s}$, this can be attributed to interaction between the three forcing components. We will refer to this as indirect effects. We define $\mathrm{Sum}\Delta$ to have explanatory value when it has the same sign as $\Delta Q_{2010s}$. We calculate the contribution of the direct effects ($\phi$) using the following equation:

$$\phi = \begin{cases} \frac{\min(\mathrm{Sum}\Delta, \Delta Q_{all})}{\max(\mathrm{Sum}\Delta, \Delta Q_{all})}, & \text{if } \mathrm{sign}(\mathrm{Sum}\Delta) = \mathrm{sign}(\Delta Q_{all}) \\ 0, & \text{if } \mathrm{sign}(\mathrm{Sum}\Delta) \neq \mathrm{sign}(\Delta Q_{all}) \end{cases} \tag{2}$$

This value can then be used to calculate the relative (direct) contribution of each forcing variable, using the following equation:

$$\phi_x = \frac{\mathrm{abs}(\Delta Q_x)}{(\mathrm{abs}(\Delta Q_P) + \mathrm{abs}(\Delta Q_{T\ snow}) + \mathrm{abs}(\Delta Q_{T\ evap}))} \cdot \phi \tag{3}$$

where $\Delta Q_x$ should be replaced by $\Delta Q_P$, $\Delta Q_{T\ snow}$, or $\Delta Q_{T\ evap}$.

[Figure]

**Figure 1.** Conceptualization of the forcing swap experiment, showing the different simulations (rows) and steps in the analysis (columns). The different forcing variables are visualized as colored blocks, where the solid and dashed boxes indicate forcing data from the 1980s and the 2010s, respectively.

---

## Author Comment (AC3) · 19 Nov 2020

We would like to thank Anonymous Referee #3 (AR3) to write this review. Below, we will reply to the points made by AR3, with the comments from AR3 in black, our response in blue.

The paper "Seasonal discharge response to temperature-driven changes in evaporation and snow processes" aims at testing the hypotheses that both seasonal changes in snowmelt and enhanced evaporation can exacerbate low flows, and that changes will increase with temperature under realistic warming. I think this study fits very well with the scope of ESD journal and it addresses an important and timely topic. However, the authors should first address some major issues before this manuscript can be considered for possible publication.

We would like to thank the reviewer for the time and the valuable suggestions. We are happy to read that the reviewer considers the topic we address timely and important. We agree with the issues raised by the reviewer and believe that addressing them will improve the quality of the manuscript.

My first concern is related to the structure and readability of the paper. I found the description of the method really poor, fragmented, and the results section is a mix of both findings and methods. Overall, I believe that this paper structure makes the manuscript confusing and difficult to read. Why not add a "case study" and "Experimental setup" sections before describing the results? Also, the authors described figure 2.a and then they jumped to figure 3, while the remaining part of figure 2 is described only after. A solution could be to split figure 2 into different ones. Try to be more consistent.

We agree with the point, and it was also brought up by the two other reviewers. We will extend the methods section, and add a better description on the experimental setup. We agree to split figure 2 into two separate figures, and will use another figure to visualize the forcing-swapping (see below).

My second concern regards the swapped method introduced in this study to understand how the individual forcing variables affect the hydrological cycle. I found this approach quite atypical, and an adequate justification should be included for why such an approach is employed. I am not deeply familiar with this swapped approach, but there is no reasoning provided for why such an approach should be preferred over other statistical approaches.

We hope that the new description of our approach (see below) removes your concerns. If not, please let us know how to improve this.

Another serious issue is the model structure. Several major hydraulic works and flood control measures were constructed over the years in the Rhine basin, strongly modifying its hydrological cycle and flood responses. How are those structures included in your distributed efficient hydrological model? This can be a major issue as major hydraulic works may have a higher influence than the forcing variables analyzed in this paper, thus compromising the findings and conclusions of the study.

It is correct that the Rhine model does contain management structures. Despite this, we show that the model is able to correctly simulate the discharge across several subbasins. This performance will likely reduce when looking only at hydrological extremes, which we are not investigating in this study. In this study, we investigate the hydrological response to changes in temperature, but how river management will change under changing condition is also a large unknown.

Linking to the structural issue of the manuscript, the authors can clearly mention in the new "Experimental setup" section that the swapped and changing-temperature approaches are meant to answer the two hypotheses of this study (see introduction). Moreover, to further improve the readability of the paper, the authors could better connect the two hypotheses of the introduction with the results summarized in the conclusions. Right now everything is there, but it takes quite some time to grasp the main take-home message.

This is a good point, and we will link the results back to the hypotheses in the conclusion section.

The authors mentioned in the conclusions that "Here we selected a resolution of 4×4 km, so we can use the ERA5 forcing data without downscaling methods (adding uncertainty and potential errors)". However, if ERA5 has a resolution of 0.25degree, how it is possible not to downscale the dataset to adapt it to a higher resolution of 4km?

This is indeed described in an ambiguous way. We did use bilinear downscaling, but argue that the scale difference is still small enough to not have to used more advanced downscaling methods. We will clarify this in the next version of the manuscript.

". . . yet these results can be interpreted for the many different basins around the globe depending on both rain- and snowfall". This sentence should be rephrased as you do not know what can occur over other basins with totally different characteristics. The results of this study cannot be generalized to other studies without proper large-scale validations. The same applies to other generalizations introduced in the conclusions. Include study limitations in the conclusion section

50     We agree that this may be misleading. We will clearly state the limitations and adapt our conclusions in the next version of the manuscript.

**1 Methods - Forcing swap**

In the first experiment, we aim to understand how each forcing variable can explain the resulting changes in discharge, and their relative importance. To perform this, we setup the experiment according to the conceptual overview presented in Fig. 1. In order to investigate how temperature influences evapotranspiration and snow processes separately, we perform model runs in which the total temperature change is splitted into temperature effects on evapotranspiration ("Changed $T_{evap}$") and snow processes ("Changed $T_{snow}$"). In addition, another run is performed with only changes in P ("Changed P"), so that these individual runs can be compared to a run where all changes in forcing are enabled (2010s forcing). The resulting simulated discharge is compared to the 1980s run, to determine the discharge change. In this way, we can evaluate the relative impact of each forcing variable on the discharge.

We sum the discharge changes of the three forcing swapped runs, to obtain $\text{Sum}\Delta$:

$$\text{Sum}\Delta = \Delta Q_P + \Delta Q_{T\ snow} + \Delta Q_{T\ evap} \tag{1}$$

where $\Delta Q_x$ represents the discharge difference of the forcing swapped simulations. We can use this $\text{Sum}\Delta$ to study how well it explains the 2010s run, by comparing it to $\Delta Q_{2010s}$. We hypothesise that when $\text{Sum}\Delta$ is equal to $\Delta Q_{2010s}$, the effect of the forcing is additive, and together explain all differents. We will refer to this as the direct effects. In the case of a discrepancy between $\text{Sum}\Delta$ and $\Delta Q_{2010s}$, this can be attributed to interaction between the three forcing components. We will refer to this as indirect effects. We define $\text{Sum}\Delta$ to have explanatory value when it has the same sign as $\Delta Q_{2010s}$. We calculate the contribution of the direct effects ($\phi$) using the following equation:

$$\phi = \begin{cases} \frac{\min(\text{Sum}\Delta, \Delta Q_{all})}{\max(\text{Sum}\Delta, \Delta Q_{all})}, & \text{if sign}(\text{Sum}\Delta) = \text{sign}(\Delta Q_{all}) \\ 0, & \text{if sign}(\text{Sum}\Delta) \neq \text{sign}(\Delta Q_{all}) \end{cases} \tag{2}$$

This value can then be used to calculate the relative (direct) contribution of each forcing variable, using the following equation:

$$\phi_x = \frac{\text{abs}(\Delta Q_x)}{(\text{abs}(\Delta Q_P) + \text{abs}(\Delta Q_{T\ snow}) + \text{abs}(\Delta Q_{T\ evap}))} \cdot \phi \tag{3}$$

where $\Delta Q_x$ should be replaced by $\Delta Q_P$, $\Delta Q_{T\ snow}$, or $\Delta Q_{T\ evap}$.

[Figure]

**Figure 1.** Conceptualization of the forcing swap experiment, showing the different simulations (rows) and steps in the analysis (columns). The different forcing variables are visualized as colored blocks, where the solid and dashed boxes indicate forcing data from the 1980s and the 2010s, respectively.

---

## Referee Report (RR1)

I do not think the increase in discharge due to increased snow melt (processes) is a major conclusion. Suggesting that it partially offsets reductions in discharge is misleading. It seems that if you added up the increases during colder parts of the year with the decreases during warmer parts, they would equal 0 and changes in snow melt (processes) would not have much impact?

Abstract: This is still confusing to understand what was done in this study. I highly recommend using the term snowmelt throughout the paper as a simple and consistent replacement for "snow processes" or "snow dynamics".

L1-6: "… how temperature-driven changes in evaporation and snow processes influence the discharge". Fourth sentence: "… observed changes could be explained by the changes induced by snow, evaporation and precipitation". I would try to make these sentences consistent. Mention specifically in the first sentence which parameters influenced discharge. Further, there are many snow processes, I would just mention the ones in your model that effect discharge (i.e. snowmelt (including glacier melt)). I am confused with attributing some changes in discharge to snow and precipitation. Snow is precipitation, so is it included in precipitation? Or do you mean liquid precipitation?

L6: Changes in precipitation explained more of observed changes in discharge than changes in snow or evaporation, but in the title and throughout the paper you focus on evaporation and snow processes. Why are changes in precipitation left out of the title and not mentioned throughout the paper when changes in snow processes and evaporation are?

L7: Higher temperatures led to earlier snowmelt (faster winter snowmelt rates) and less available snowpack to melt later in spring, when it historically melts.

L16: Need a reference for "potentially higher snowmelt rates". Perhaps you are talking about an increase in winter snowmelt rates? ("Melt trends portend widespread declines in snow water resources")

L17-18: This new sentence comes off out of place without additional context.

L20: "Water towers" were defined earlier "Mountains of the world, water towers for humanity: Typology, mapping, and global significance". Also, add "to" between important and have.

L26: Wouldn't only melted snow affect discharge?

L50: I am more of a fan of using the term "snowmelt" compared to "snow processes"

L64: Delete "upstream of the basin".

L149-150: How do you separate temperature effects on ET or snow processes? Wouldn't changing the temperature affect both processes at the same time? Are you only changing temperature during the winter/summer and attributing that to only affecting snow/ET?

L244: What do you mean by snow and evaporation are threshold processes?

L246: add "periods" between 'the' and 'during'

L248: "Partially offset" seems like a stretch for the Jan-Feb period. The black line barely deviates from the orange line.

L266: Switch "changed" to "change"

L324: Can you differentiate between snow melt and ice melt?

L330-331: Can this be explained with your data? Yes, the blue line in Fig. 5a is often barely above the dashed black line, but I would not rely on it too much. This statement is misleading if you cannot differentiate between snow and glacier melt and presents a potential false hope.

---

## Referee Report (RR2)

I think the addition of Figure S4 is very helpful for interpreting the results, but would like to see more discussion put into the 'offsetting' effects of increased snow processes throughout the majority of the year and more of an attempt to disentangle the effects from snowmelt and glacier melt. For example, in the supplementary you show decreases of snow storage (S3), snow cover (S3), and snowmelt rate (S4) with increasing temperature, but increasing glacier melt (S4) (it would be better if S3 and S4 had the same range for x-axis for easier comparison). Grouping snowmelt and glacier melt into the same term is misleading as they represent two water storages with very different residence times and thus different consequences of ecosystem vulnerability to warming scenarios.

- The results from Figure S4 imply the increase in Jan-Feb discharge (Figure 5b, 6a) is caused by increases in glacier melt, since snow melt rates decrease during this time (and all times). Is this correct? Based on the larger magnitude of glacier melt during the summer from Figure S4, I would have expected larger changes in discharge to occur in the summer (i.e. more blue pixels during May-Jun and Sep-Oct compared to Jan-Feb).
- Why do increases in discharge occur from snow processes during Jan-Feb (Figure 5, 6a) if snowmelt rate decreases during this time and glacier melt is close to 0? Can you rescale the y-axis in Figure S4 so that each plot is comparable (i.e. averaged over the entire basin)?
- Based on the results from Figure S4, the offsetting effect is solely due to increased glacier melt, as snowmelt is shown to decrease under warmer scenarios for all months. This needs to be discussed more as the water stored in glaciers is finite, thus this offsetting effect will be eliminated under warmer scenarios when the glaciers have completely melted.

L11: Please specify what an interaction of the variables means in the abstract since more than half of your results are due to the interaction and understanding the interaction is vital.

L14: When applicable, please mention whether the 'changes' are positive or negative. Switching the language to 'increases' or 'decreases' would ease in interpreting the main results. You use the terms 'changes' and 'differences' throughout the manuscript and it would be easier to interpret if you clarify if they are positive or negative changes.

L14 / 386: This is still one of my biggest concerns in regards to interpreting your results. Again, I think the term 'partially offset' is misleading as discharge clearly decreases due to increases in evaporation and the increased snow processes barely make a difference. Also, be consistent of what the low flow period refers to. Earlier in the manuscript lowflow referred to Sep-Oct, now you are including Nov too. I feel that the main take-away from the second experiment (Figure 5), which should be emphasized more in the abstract and conclusion than the offsetting from snow processes, is that simulated discharge is lower in every month and primarily caused by increases in evaporation.

Further, since these results do not distinguish between snow melt and glacier melt, this is problematic if most of this offset is due to glacier melt (as indicated by S4), which is a finite supply of water, thus will eventually disappear given enough warming. I think you should clarify that this 'offsetting' will not occur in future warmer scenarios when the glaciers are depleted and more precipitation falls as rain compared to snow.

L201: Rewrite "For example, temperature and precipitation are linked as the type of precipitation (rain or snow) is dependent on temperature.". It is still confusing what the interaction of the forcing components refers to.

L242: change "in stead" to "instead".

L274: The term 'snow processes' is still confusing to me. Why is 'liquid precipitation' in parentheses following 'snow processes' (again in L361). Are you referring to snowmelt and not rain?

L293/301: Please include the complete figure reference (i.e. Figure 5b & 5d instead of panel b and d). Further, please add more references to the figures when possible for justifying your statements. For instance, on L302 you write "Substantial influence of rooting depth on the evaporation simulation is visible", but nothing is 'visible' if there is no reference to a figure or table for the reader.

L349: Why switch "increased" to "exacerbated", they have opposite meanings here?

L382-384: Please specify whether the 'differences' or 'changes' are positive or negative when applicable here and throughout the manuscript.

---

## Author Response (AR2)

**List of all changes**

- We have rewritten the abstract to better summarize the different parts of our study, and have clarified several confusing sentences.

- Throughout the manuscript, we have clarified our usage of the term "snow processes" as it was not always clear what processes we were referring to. Additionally, other confusing sentences have been clarified as well.

- We added a brief description and the equations of the snow module of the dS2 model, as this helps to understand the results.

- We improved the structure of the manuscript by moving text to their appropriate sections (from results to methods and discussion).

- We added a figure in the supplement, which shows the melt rate from snow and ice, to support the claims made in the main manuscript.

We would like to thank Anonymous Referee #1 (AR1) to write this review. Below, we will reply to the points made by AR1, with the comments from AR1 in black, and our response in blue.

**General comment**

The manuscript considerably improved compared to the other version I saw. The text is structured better and the model set-up explained better. In the following, some more minor issues that should be addressed before it is considered for publication. When specifying line numbers, I refer to the marked-up manuscript.

   Thank you for reading and reviewing our improved version of the manuscript. We understand the issues raised, and comment with our solutions below.

**Specific comments:**

   – Structure text: Even though the structure of the text is better now, one more revision is necessary, in my opinion. There still are some elements in the result section that should be moved to the method part. Please discuss your results in the 'Discussion'. Still almost all the discussion is in the result section. Add sub-section to the discussion (preferably the same that you use in the results).

Thanks for this suggestion. We have moved the relevant paragraphs from the results to either the methods or the discussion. We do believe however, that some interpretation of the results is necessary to understand the results and the following analyses.

   – Line 8: Please explain (what you mean by combination of these variables.

   Here we refer to the interaction between e.g. precipitation and temperature. If temperature drops below 0°C, precipitation
will fall as snow, and will hence have a different effect on the resulting discharge. We have revised the abstract to clarify this.

   – Line 19-20: Suggestions:: write directly 'less precipitation and lower snowmelt rates' and remove 'potentially higher snowmelt rates' and 'surprisingly'.

   We have clarified the sentence, and move the reference to Musselman et al. (2017) to the discussion.

– Line 61: expected > potential

   Thanks for the suggestion, we have changed this.

   – Fig. 1: You detect less precipitation in the second time window. How do you explain this? Before you mention the intensification of the hydrological cycle. Should this cause more precipitation? Or is it just decadal variability?

   We attribute this to decadal variation, as periods of 10 years are to short to make qualitative assessments about changes
in climate.

   – Line 125 ff: The model does not include lakes or reservoirs, as far as I understood. This should be mentioned here and discussed later, as this is important for low flow situations.

   This is correct. We already discuss this in the discussion section, but we added this to the initial model description as well.

– Line 139: You run in hourly resolution, right? How does this work when you have a degree-day approach? What glaciers do you simulate? How does the model know that there is ice to simulate in a pixel? In case the snow and glacier modules are new, their implementation should be explained a bit better, I think.

We do indeed run the model on a hourly timestep, and the snow module is corrected for this. This means that the melt factors are defined as $\mathrm{mm\,°C^{-1}h^{-1}}$. We defined several pixels as glacier pixels, which are therefore fixed in space. These pixels also receive the glacier melt factors.

– Line 156: Please provide references of other studies using this swapping approach and discuss (dis-)advantages better.

To our knowledge, this approach has not yet been applied yet. We have added a statement about the pros and cons of this method.

– Line 171: Please explain better what you mean by 'interaction between the three forcing components. This is an important
point of your study and should be explained in more detail, I think.

We have added a description of what we mean with the interactions, including the temperature-precipitation example.

– Line 180: You add 0.5 ° to you hourly temperature time series? If yes, please discuss possible shortcomings of this approach.

This is correct, and we have added a downside of this approach.

– Line 198-199: This information needs to be in the method section.

Thanks for the suggestion, we have moved it to the method section.

– Fig. 3: How do you determine that differences are significant? Significant differences in median values?

We compare the discharge values from the 1980s with discharge values from the 2010s, grouped per month. We perform an independent t-test using these two groups of data, to determine the p-value for each month.

– LIne 205-209: Move this information to the section on data and methods. Do not introduce new data of validation approaches in the result section.

We have fixed this, and the validation procedure is now explained in the method section.

– Line 210: Please explain why you use this catchment.

This catchment was selected as the data is of high quality and at a high temporal resolution, which matches the model
timestep. We have added this information to the manuscript.

– Line 240: 'ice' Glacier melt during March and May? Can you check whether it really is snow + ice or only snowmelt?

It was slightly wrong written: it is a combination of more melt from the glaciers (due to the increased average temperatures), and more direct runoff as more precipitation is falling as rain in stead of snow.

– Fig. 4: To me it is confusing why there are two ylab next to panel d. Please move to correct panels.

We use two labels as the two panels (e and f) use different units (cm and mm, respectively), similarly to panel d. We have clarified this in the caption.

– Line 246-249: I do not understand this explanation. Why are 'storage conditions' different between the months of March and April?

We have improved the explanation of this, as the references to the periods were confusing

– Fig. 3: Please provide an explanation for the almost sinusoidal variation in your result figures.

This sinusoidal variation is largely driven by the change in precipitation, which is simply due to the differences between the decades.

– Line 267-269: How is it possible that you get opposite results in your experiments for Jan-Feb with regard to changes in snowmelt contribution. I am surprised as well. Please explain better

An explanation has been added to describe the cause of this increase.

– Line 269 and Line 351: 'snow processes': Please specify what you mean by snow processes. Is it accumulation, snowmelt rates, timing,...

We have added more explicit descriptions of snow processes where necessary.

– Line 285: Move to discussion. Please do not discuss you results in the result section. Discuss in Discussion.

This paragraph describes the results shown in Fig. 6b, but requires some explanation to understand the results.

– Line 297: 'Over these periods of 10 years, most interannual variability is average out'

We have added a statement about the remaining present decadal variation.

– Line 348: I still have troubles to understand how you get to this number of 19% for precipitation. Please explain in the method section.

It is taken directly from Fig. 3f $(0.45 - 0.26)$, where the mean values of panel e are presented. The way the contribution of each forcing variable is calculated is shown in Equation 11.

– Line 351: It sounds like there is more snowmelt with higher temperatures in Sep-Oct. Is this what you want to say? Isn't it the opposite? Higher temperatures result in a reduction of snow accumulation and a reduced build up of a snow cover. Rainfall is liquid instead of solid and hence more runoff?

This is indeed correct, and we have clarified this (more liquid precipitation, but enhanced melt from the glaciers).

– Line 264: Where do you snow enhanced snowmelt in you study? I do not see a figure depicting changes in snowmelt (rates) in you study.

Assuming you refer to Line 364, we have clarified this sentence. Additionally, we have added timeseries to show the melt rates for both snow and glaciers in the supplement (where also timeseries of the snowpack and -cover can be found).

We want to thank Anonymous Referee #2 (AR2) for taking the time to review the improved version of the manuscript. The comments from AR2 are presented in black, and our response in blue.

**Comments**

– I do not think the increase in discharge due to increased snow melt (processes) is a major conclusion. Suggesting that it partially offsets reductions in discharge is misleading. It seems that if you added up the increases during colder parts of the year with the decreases during warmer parts, they would equal 0 and changes in snow melt (processes) would not have much impact?

    This is correct when taking the yearly average value. Our conclusion that this is the case refers, however, to specific
periods. We will clarify this.

  – Abstract: This is still confusing to understand what was done in this study. I highly recommend using the term snowmelt throughout the paper as a simple and consistent replacement for "snow processes" or "snow dynamics".

    We have rewritten the abstract to more clearly describe the two experiments, and replaced the term "snow processes" with "snowfall and melt from snow and ice," as both are important.

– L1-6: "... how temperature-driven changes in evaporation and snow processes influence the discharge". Fourth sentence: "... observed changes could be explained by the changes induced by snow, evaporation and precipitation". I would try to make these sentences consistent. Mention specifically in the first sentence which parameters influenced discharge. Further, there are many snow processes, I would just mention the ones in your model that effect discharge (i.e. snowmelt (including glacier melt)). I am confused with attributing some changes in discharge to snow and precipitation. Snow is
precipitation, so is it included in precipitation? Or do you mean liquid precipitation?

    The three variables we changed in the first experiment are evaporation, temperature (affecting only snowfall, and melt from snow and ice), and precipitation. Since the focus of our manuscript is on effect of temperature changes, we focus on evaporation, snowfall and melt. Precipitation input is defined as total precipitation, the model divides between rain and snow based on the temperature timeseries. We have clarified this in the newest version of the manuscript.

– L6: Changes in precipitation explained more of observed changes in discharge than changes in snow or evaporation, but in the title and throughout the paper you focus on evaporation and snow processes. Why are changes in precipitation left out of the title and not mentioned throughout the paper when changes in snow processes and evaporation are?

    The focus of the paper is on temperature driven changes in evaporation and snowfall and melt. When comparing two decades, changes in precipitation become indeed very important. However, our results show that in this decadal compar-
ison the temperature driven changes are larger (26%) than the precipitation driven changes (19%). We have clarified this in the next version of the manuscript.

  – L7: Higher temperatures led to earlier snowmelt (faster winter snowmelt rates) and less available snowpack to melt later in spring, when it historically melts.

    Thanks for this suggestion, we have clarified the sentence.

– L16: Need a reference for "potentially higher snowmelt rates". Perhaps you are talking about an increase in winter snowmelt rates? ("Melt trends portend widespread declines in snow water resources")

    Higher temperatures means that more energy is available to melting. We have rewritten this section and moved the confusing reference to Musselman et al. (2017) to the discussion.

  – L17-18: This new sentence comes off out of place without additional context.

We decided to remove this sentence, as we cover this work in the discussion.

– L20: "Water towers" were defined earlier "Mountains of the world, water towers for humanity: Typology, mapping, and global significance". Also, add "to" between important and have.

Thanks for this correction, we have fixed it.

– L26: Wouldn't only melted snow affect discharge?

Whether precipitation falls as rain or snow also affects discharge. Here we wanted to explain the term "snow processes" but we understand this is still confusing.

– L50: I am more of a fan of using the term "snowmelt" compared to "snow processes"

We understand the ambiguity of the term "snow processes" and we will therefore replace this term in the text with the exact processes we refer to (snowfall, snowmelt and melt from glaciers).

– L64: Delete "upstream of the basin".

Thanks for the correction.

– L149-150: How do you separate temperature effects on ET or snow processes? Wouldn't changing the temperature affect both processes at the same time? Are you only changing temperature during the winter/summer and attributing that to only affecting snow/ET?

The model uses separate inputs for evaporation and temperature (for snowfall and melt). This way, we provided time-series where only the evaporation is changed, and timeseries where only the temperature is changed. This is also how we setup the increased temperature experiment.

– L244: What do you mean by snow and evaporation are threshold processes?

Snowfall and melt are dependent on temperature, where temperature acts as a threshold to "activate" these processes.
Evaporation is dependent on (amongst others) soil moisture, where insufficient soil moisture acts as a threshold to reduce actual evaporation. We have clarified this in the newest version of the manuscript.

– L246: add "periods" between 'the' and 'during'

We have fixed this.

– L248: "Partially offset" seems like a stretch for the Jan-Feb period. The black line barely deviates from the orange line.

There is a small deviation, yet it is still present. We will rephrase this sentence.

– L266: Switch "changed" to "change"

Thanks for the correction, we have fixed this.

– L324: Can you differentiate between snow melt and ice melt?

We have added average melt rates for snow and ice in the supplement, to better depict how higher temperatures affect
melt from snow and ice.

– L330-331: Can this be explained with your data? Yes, the blue line in Fig. 5a is often barely above the dashed black line, but I would not rely on it too much. This statement is misleading if you cannot differentiate between snow and glacier melt and presents a potential false hope.

We defined the term snow processes to include snowfall, snowmelt and melt from glaciers. Despite it being a small change (for the majority of the year, both the purple line is slightly above the dashed black line, and the brown line is slightly above the orange line), it still slightly offsets the negative change induced by the enhanced evaporation. We understand the ambiguity around the snow processes term, and this will hopefully be fixed by explicitly stating which processes we refer to.

---

## Author Response (AR3)

**List of relevant changes**

- We have clarified the usage of the term snow processes, as it was previously not sufficiently clear that the type of precipitation (rain or snow) was also included in the model and is essential for the correct interpretation of the results.

- We have slightly rewritten the abstract and conclusion to better represent the main findings of the manuscript.

- We have added examples on the forcing interactions that our forcing swap approach cannot capture.

- A new figure was added to the supplement, replacing the snow storage and melt figures in the previous version. This figure adds a precipitation time series, showing the amount of precipitation falling as snow or rain. Additionally, we ensured all panels have the same x-axis, and have corrected all relevant values to basin average mm h$^{-1}$ units.

- We added a new paragraph to the discussion to describe the limitations of the way we simulate glaciers.

**Reply to review**

We would like to thank Anonymous Referee #2 (AR2) again for reading our latest version of the manuscript. We understand the confusion, which is hopefully clarified with a revised graph in the supplement. Below, we will reply to the points made by AR2, with the comments from AR2 in black, and our response in blue.

- I think the addition of Figure S4 is very helpful for interpreting the results, but would like to see more discussion put into the 'offsetting' effects of increased snow processes throughout the majority of the year and more of an attempt to disentangle the effects from snowmelt and glacier melt. For example, in the supplementary you show decreases of snow storage (S3), snow cover (S3), and snowmelt rate (S4) with increasing temperature, but increasing glacier melt (S4) (it would be better if S3 and S4 had the same range for x-axis for easier comparison). Grouping snowmelt and glacier melt into the same term is misleading as they represent two water storages with very different residence times and thus different consequences of ecosystem vulnerability to warming scenarios.

  Thanks for this suggestion. We have combined the two snow-related figures in the supplement into a single figure, to better depict how changes in type of precipitation and melt affect different components of the hydrological cycle. This new figure shows the amount of total precipitation, and the amount of this precipitation that is falling as snow. More rainfall results in more direct discharge, and we have stated this more clearly in the newest version of the manuscript. Additionally, we understand the concerns about the grouping of melt from both snow and glaciers. This figure in the supplement helps to depict the response of these two factors better, but we have described this more carefully in the latest version of the manuscript as well.

- The results from Figure S4 imply the increase in Jan-Feb discharge (Figure 5b, 6a) is caused by increases in glacier melt, since snow melt rates decrease during this time (and all times). Is this correct? Based on the larger magnitude of glacier melt during the summer from Figure S4, I would have expected larger changes in discharge to occur in the summer (i.e. more blue pixels during May-Jun and Sep-Oct compared to Jan-Feb).

  This is partly correct, but is largely caused by the change in type of precipitation. We hope the new panel, and better descriptions in the manuscript clarify this. Assuming the blue pixels refer to Figure 6: these blue pixels are the glaciers and are the pixels responsible for the slight increase in discharge.

- Why do increases in discharge occur from snow processes during Jan-Feb (Figure 5, 6a) if snowmelt rate decreases during this time and glacier melt is close to 0? Can you rescale the y-axis in Figure S4 so that each plot is comparable (i.e. averaged over the entire basin)?

  Thanks for the suggestion on the y-axis, we have implemented this. This increase in caused by more direct runoff (liquid versus solid precipitation). We have stated this more clearly in the manuscript.

– Based on the results from Figure S4, the offsetting effect is solely due to increased glacier melt, as snowmelt is shown to decrease under warmer scenarios for all months. This needs to be discussed more as the water stored in glaciers is finite, thus this offsetting effect will be eliminated under warmer scenarios when the glaciers have completely melted.

*This is correct, and we have added this to the discussion.*

– L11: Please specify what an interaction of the variables means in the abstract since more than half of your results are due to the interaction and understanding the interaction is vital.

*We have added two examples to describe the interaction of the variables.*

– L14: When applicable, please mention whether the 'changes' are positive or negative. Switching the language to 'increases' or 'decreases' would ease in interpreting the main results. You use the terms 'changes' and 'differences' throughout the manuscript and it would be easier to interpret if you clarify if they are positive or negative changes.

*In some cases, the changes imply both the negative and positive changes. We have clarified these sentences.*

– L14 / 386: This is still one of my biggest concerns in regards to interpreting your results. Again, I think the term 'partially offset' is misleading as discharge clearly decreases due to increases in evaporation and the increased snow processes barely make a difference. Also, be consistent of what the low flow period refers to. Earlier in the manuscript low flow referred to Sep-Oct, now you are including Nov too. I feel that the main take-away from the second experiment (Figure 5), which should be emphasized more in the abstract and conclusion than the offsetting from snow processes, is that simulated discharge is lower in every month and primarily caused by increases in evaporation.

*We understand how the term "partially offset" could be misleading. We have rephrased this throughout the manuscript to be better in line with the results.*

– Further, since these results do not distinguish between snow melt and glacier melt, this is problematic if most of this offset is due to glacier melt (as indicated by S4), which is a finite supply of water, thus will eventually disappear given enough warming. I think you should clarify that this 'offsetting' will not occur in future warmer scenarios when the glaciers are depleted and more precipitation falls as rain compared to snow.

*We have added this to the discussion, as this is indeed a valid point.*

– L201: Rewrite "For example, temperature and precipitation are linked as the type of precipitation (rain or snow) is dependent on temperature.". It is still confusing what the interaction of the forcing components refers to.

*We have added a more detailed example on what we mean with these interactions. Hopefully this will remove the confusion.*

– L242: change "in stead" to "instead".

*Thanks for this correction.*

– L274: The term 'snow processes' is still confusing to me. Why is 'liquid precipitation' in parentheses following 'snow processes' (again in L361). Are you referring to snowmelt and not rain?

*When we change the temperature, this will also affect type of precipitation that falls (see also the new example mentioned two comments above). Additionally, with the addition of the snowfall panel in the supplement, we hope that this clarifies that we include the type of precipitation (rain or snow) in the term "snow processes".*

– L293/301: Please include the complete figure reference (i.e. Figure 5b & 5d instead of panel b and d). Further, please add more references to the figures when possible for justifying your statements. For instance, on L302 you write "Substantial influence of rooting depth on the evaporation simulation is visible", but nothing is 'visible' if there is no reference to a figure or table for the reader.

80    Thanks for this suggestion, we have fixed this.

– L349: Why switch "increased" to "exacerbated", they have opposite meanings here?

This is indeed incorrect, and we have fixed and clarified this sentence.

– L382-384: Please specify whether the 'differences' or 'changes' are positive or negative when applicable here and throughout the manuscript.

85    We have clarified this throughout the manuscript.